# Nfat/calcineurin signaling promotes oligodendrocyte differentiation and myelination by transcription factor network tuning

Matthias Weider[1], Laura Julia Starost[2,3], Katharina Groll[2], Melanie Küspert[1], Elisabeth Sock[1], Miriam Wedel[1], Franziska Fröb[1], Christian Schmitt[1], Tina Baroti[1], Anna C. Hartwig[1], Simone Hillgärtner[1], Sandra Piefke[1], Tanja Fadler[1], Marc Ehrlich[2,3], Corinna Ehlert[2], Martin Stehling[3], Stefanie Albrecht[2], Ammar Jabali[2], Hans R. Schöler[3], Jürgen Winkler[4], Tanja Kuhlmann[2] & Michael Wegner [1]

Oligodendrocytes produce myelin for rapid transmission and saltatory conduction of action potentials in the vertebrate central nervous system. Activation of the myelination program requires several transcription factors including Sox10, Olig2, and Nkx2.2. Functional interactions among them are poorly understood and important components of the regulatory network are still unknown. Here, we identify Nfat proteins as Sox10 targets and regulators of oligodendroglial differentiation in rodents and humans. Overall levels and nuclear fraction increase during differentiation. Inhibition of Nfat activity impedes oligodendrocyte differentiation in vitro and in vivo. On a molecular level, Nfat proteins cooperate with Sox10 to relieve reciprocal repression of Olig2 and Nkx2.2 as precondition for oligodendroglial differentiation and myelination. As Nfat activity depends on calcium-dependent activation of calcineurin signaling, regulatory network and oligodendroglial differentiation become sensitive to calcium signals. NFAT proteins are also detected in human oligodendrocytes, downregulated in active multiple sclerosis lesions and thus likely relevant in demyelinating disease.

[1] Institut für Biochemie, Emil-Fischer-Zentrum, Friedrich-Alexander-Universität Erlangen-Nürnberg, D-91054 Erlangen, Germany. [2] Institute of Neuropathology, University Hospital Münster, D-48149 Münster, Germany. [3] Department of Cell and Developmental Biology, Max Planck Institute for Molecular Biomedicine, D-48149 Münster, Germany. [4] Department of Molecular Neurology, University Hospital Erlangen Friedrich-Alexander-Universität Erlangen-Nürnberg, D-91054 Erlangen, Germany. These authors contributed equally: Matthias Weider, Laura Julia Starost and Katharina Groll. These authors jointly supervised this work: Tanja Kuhlmann, Michael Wegner. Correspondence and requests for materials should be addressed to T.K. (email: Tanja.Kuhlmann@ukmuenster.de) or to M.W. (email: michael.wegner@fau.de)

Developmental processes such as generation and terminal differentiation of oligodendrocytes as well as myelination are governed by complex gene regulatory networks that integrate extrinsic and intrinsic stimuli into a coordinate response. A detailed knowledge of the interactions within the network is not only essential for understanding developmental myelination but also for establishing novel approaches for the treatment of demyelinating diseases, such as multiple sclerosis (MS), in which the formation of new myelin sheaths (i.e., remyelination) after a demyelinating event is frequently impaired due to a failure of oligodendrocyte differentiation[1–3].

Several central components of the regulatory network in oligodendrocytes have been identified over the years and include the transcription factors Olig2, Sox10, Nkx2.2, and Myrf as major determinants of oligodendroglial differentiation and myelination[4]. Olig2 is already expressed at the time of oligodendroglial specification and triggers the induction of Sox10 as a direct target gene[5–9]. Once induced, Sox10 contributes to maintenance of Olig2 expression in a positive feedback loop by directly activating an upstream enhancer (OLE, in particular the distal OLEa part) of the *Olig2* gene[10]. Sox10 also stimulates Nkx2.2 expression and induces Myrf prior to the onset of terminal differentiation[11, 12].

The essential co-expression of Olig2 and Nkx2.2 in differentiating oligodendrocytes[5, 6, 8, 9] contrasts with the mutually exclusive expression pattern of these two factors at earlier times. When oligodendrocyte precursor cells (OPCs) are generated and specified from neuroepithelial cells, Olig2, and Nkx2.2 are expressed in adjacent domains of the ventral ventricular zone of the central nervous system (CNS) and cross-repress each other[13–15]. Terminal differentiation of oligodendrocytes and myelination thus require this cross-repression to be relieved.

Many more regulatory network components and interactions among them must exist to explain network activity and its changes upon extrinsic signals. Especially the identification of regulators that respond to extracellular signals, and their integration into the regulatory network are of utmost importance to explain how the influence of intrinsic and extrinsic factors on oligodendroglial development and myelination is coordinated.

Nfat proteins are such regulators, as their activity depends on increases in intracellular calcium levels and is mediated by the calcium-dependent phosphatase calcineurin and calcineurin-dependent dephosphorylation events[16]. Nfat activation often goes along with a translocation from cytosol to nucleus.

Here we identify Nfat proteins as crucial and so far unknown regulators of oligodendrocyte differentiation and integrate them into the oligodendroglial gene regulatory network. We show that the concerted action of Sox10 and Nfat proteins allows cross-repression of Olig2 and Nkx2.2 to be relieved and both proteins to be co-expressed as a precondition for oligodendrocyte differentiation.

## Results

### Nfat proteins promote rodent oligodendrocyte differentiation.

The small molecule 11R-VIVIT (VIVIT) disrupts calcineurin binding to Nfat proteins and inhibits Nfat activation. At 1 μM, VIVIT did not affect viability of mouse oligodendroglial cells (Suppl. Fig. 1a). Effects on proliferation were also minor as judged from BrdU incorporation studies of OPC cultures kept for 24 or 48 h in the absence or presence of 1 μM VIVIT (Suppl. Fig. 1b). When added to oligodendroglial cultures kept under differentiating conditions for 48 h, VIVIT dramatically reduced the number of Mbp-positive oligodendrocytes and *Mbp* transcript levels (Fig. 1a–c). A comparable decrease in Mbp-expressing cells was also detected following incubation of cultured rat oligodendroglial cells with the general calcineurin inhibitor FK506/tacrolimus (Suppl. Fig. 1c, d). In line with a function in oligodendrocyte differentiation, a tdTomato reporter under control of a Nfat-sensitive promoter preferentially segregated to Mbp-positive cells in oligodendroglial cultures (Suppl. Fig. 1e).

Reduction in *Mbp* transcripts and Mbp-positive cells was highest when VIVIT was continuously present (Fig. 1a–c). If applied for only 24 h, downregulation of *Mbp* was much more pronounced when VIVIT was present during the first 24 h as compared to the second 24 h of differentiation (Fig. 1a–c). Similar results were observed for *Mag* and *Plp1* (Suppl. Fig. 1f). This argues that calcineurin-dependent Nfat activation is particularly important during the initial stages of differentiation.

By treating murine cerebellar slice cultures with VIVIT for 12 days, we assessed the effect of Nfat inhibition on myelination (Fig. 1d–f). Visual inspection revealed a reduction of neurofilament- (Nefl-) positive axons wrapped by Mbp-positive sheaths in the presence of VIVIT (Fig. 1d). This was confirmed by quantification, as the fraction of the Nefl-positive area with Mbp co-stain and the *Nefl*-normalized amounts of *Mbp* transcripts were both significantly decreased following VIVIT treatment (Fig. 1e, f).

Oligodendroglial differentiation and myelin gene expression were also reduced after inhibition of calcineurin signaling in cortical slices prepared from newborn mice (Suppl. Fig. 1g, h). After 7 days of culture in the presence of FK506, Mbp expression was only half as high as under control conditions without obvious changes in the overall number of Sox10-expressing oligodendroglial cells.

Using conditional mouse mutagenesis, we also studied the role of calcineurin signaling in vivo. For this purpose, we generated mice in which a floxed calcineurin *CnB1/Ppp3r1* allele[17] was combined with different Cre drivers. Available Cre drivers were also active outside the CNS and led to additional widespread *CnB1* deletion in the neural crest. The ensuing developmental disturbances caused perinatal death so that consequences of *CnB1* deletion on oligodendrocyte differentiation could only be studied until birth.

A *Sox10::Cre* transgene was employed to target oligodendroglial cells immediately after their specification from ventricular zone neuroepithelial precursors[18, 19]. In the resulting *CnB1$^{\Delta Sox10}$* mice, distribution and number of Olig2-positive oligodendroglial cells in the spinal cord of 18.5 dpc old embryos were indistinguishable from the wildtype (Fig. 2a, g). At this stage, most of these cells expressed the OPC marker Pdgfra in both genotypes (Fig. 2b, h). In contrast, the number of Nkx2.2-positive promyelinating oligodendrocytes in the spinal cord of *CnB1$^{\Delta Sox10}$* mice was reduced to half the wildtype levels (Fig. 2c, i). Induction of Myrf as the central transcriptional regulator of the myelination program was similarly decreased in mutant spinal cords (Fig. 2d, j). This resulted in substantially reduced expression of myelin genes such as *Mbp* and *Plp1* (Fig. 2e, f, k, l). In contrast, oligodendroglial proliferation and apoptosis remained unchanged as compared to the wildtype (Fig. 2m, n). We conclude from these findings that calcineurin signaling is required for proper oligodendroglial differentiation and likely acts at the promyelinating stage.

We additionally deleted *CnB1* by *Cnp1::Cre*[20] and obtained comparable results in the resulting *CnB1$^{\Delta Cnp1}$* mice (Suppl. Fig. 2a–n). Considering that this Cre becomes active considerably later than *Sox10::Cre*[21] it is reasonable to conclude that calcineurin signaling is dispensable in OPCs for the most part of their lineage progression and becomes physiologically relevant in the early stages of oligodendroglial differentiation.

**Nfatc2 is active in differentiating oligodendrocytes**. Of the four calcium-regulated Nfat genes *Nfatc1*, *Nfatc2*, and *Nfatc3* were detected as transcripts in cultured mouse oligodendroglial cells by qrtPCR, whereas *Nfatc4* transcripts were absent (Fig. 3a). Relative amounts of the Nfats can be inferred from published RNA-Seq data and point to a particularly prominent *Nfatc2* expression[22]

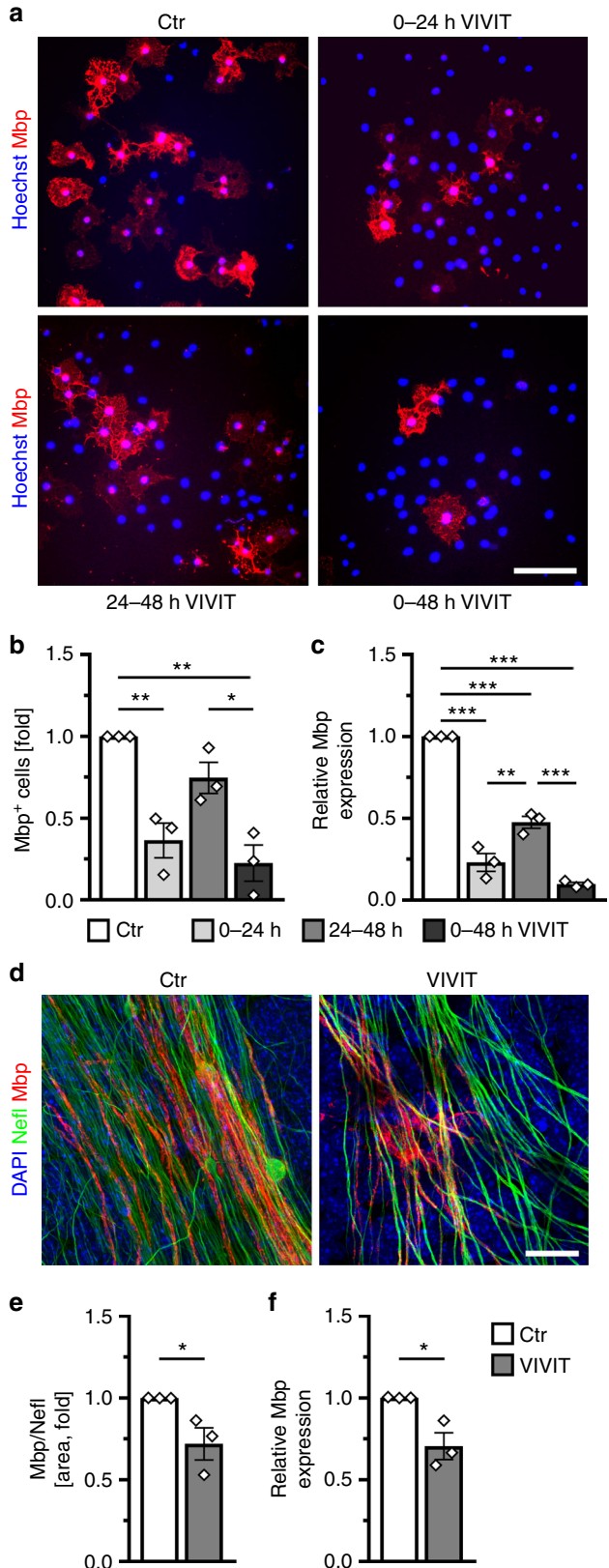

(Suppl. Fig. 3a). Expression levels furthermore increased for all detected family members during oligodendroglial differentiation in culture as evident from qrtPCR analysis (Fig. 3a).

Nfatc1, Nfatc2, and Nfatc3 were also detected as proteins by immunocytochemistry (Fig. 3b). In differentiating mouse oligodendrocytes Nfatc1 is exclusively nuclear, whereas Nfatc2 and Nfatc3 are present in nucleus and cytoplasm. The fraction of Nfatc2 with nuclear localization increases upon oligodendroglial differentiation (Suppl. Fig. 3b). Whereas intensity of the Nfatc2 signal in cultured OPCs was higher in the cytoplasm than in the nucleus, differentiating cells exhibited similar signal intensities in nucleus and cytoplasm (Suppl. Fig. 3c). Increased nuclear translocation was also observed for a Nfatc2-eYFP fusion protein[23] after transfection of the oligodendroglial Oli-neu cell line with a corresponding expression construct and differentiation of these cells by dibutyryl-cAMP (Suppl. Fig. 3d).

Nuclear Nfatc2 amounts also increased after a rise of intracellular calcium levels and calcineurin activation. Addition of ionomycin to OPC cultures led to nuclear enrichment of endogenous Nfatc2 and, after transfection of a corresponding expression plasmid, a Nfatc2-eYFP fusion protein (Fig. 3c, d). This suggests that increased Nfatc2 translocation into the nucleus during oligodendroglial differentiation is a consequence of active calcineurin signaling.

As *Nfatc2* exhibited the highest expression of all Nfats in rodent oligodendroglial cells and as nuclear localization increased with differentiation, we focused on this family member. To understand how *Nfatc2* expression is regulated at the transcriptional level in oligodendroglial cells, we made use of the oligodendroglial precursor cell line Oln93[24] and clonal variants

**Fig. 1** Nfat/calcineurin signaling is required for oligodendroglial differentiation in culture. **a–c** Analysis of myelin gene expression in primary mouse oligodendroglial cells cultured for 48 h under differentiating conditions in the absence (Ctr, open bar) or presence of 1 μM VIVIT (VIVIT, grey bars). Incubation with VIVIT was restricted to the first 24 h (light grey bars) or second 24 h (grey bars) of incubation or throughout the whole cultivation period (dark grey bars). Cultures were stained with antibodies directed against Mbp (red) and counterstained with Hoechst (blue). Scale bar, 50 μm (**a**). From immunocytochemical stainings the fraction of Mbp-positive cells was determined (**b**) (*n* = 3). The relative number of Mbp-positive cells present under control conditions was arbitrarily set to 1 and used to normalize in pairwise fashion (values: 1 for control conditions, 0.37 ± 0.11 for VIVIT treatment during the first 24 h, 0.75 ± 0.09 for VIVIT treatment during the second 24 h, 0.23 ± 0.11 for 48 h VIVIT treatment). RNA from these cultures was also used to perform qrtPCR and determine *Mbp* levels (**c**) (*n* = 3). The amount of *Mbp* transcripts present after 48 h under control conditions was arbitrarily set to 1 and used to normalize (values: 1 for control conditions, 0.23 ± 0.10 for VIVIT treatment during the first 24 h, 0.48 ± 0.06 for VIVIT treatment during the second 24 h, 0.10 ± 0.02 for 48 h VIVIT treatment). **d–f** Analysis of myelination in cerebellar slices of newborn mice after 12 days of culture in the absence (Ctr, open bars) and presence (VIVIT, grey bars) of 1 μM VIVIT (*n* = 3). Cultures were stained (**d**) with antibodies directed against Mbp (red) and Neurofilament L (Nefl, green) and counterstained with DAPI (blue). Scale bar, 20 μm. The extent of myelination was assessed by determining the Nefl-positive area co-stained with Mbp (**e**). The Mbp/Nefl ratio under control conditions was arbitrarily set to 1 and used to normalize in pairwise fashion (values: 1 for control conditions, 0.72 ± 0.09 for VIVIT treatment). RNA prepared from these cultures was used to determine *Mbp* and *Nefl* expression (**f**). The amount of *Mbp* relative to *Nefl* transcripts under control conditions was arbitrarily set to 1 (values: 1 for control conditions, 0.71 ± 0.08 for VIVIT treatment). Statistical significance was determined by Bonferroni-corrected one-way ANOVA in (**b**) and (**c**) and two-tailed Student's *t*-test in (**e**) and (**f**) (*$P \leq 0.05$; **$P \leq 0.01$; ***$P \leq 0.001$)

in which *Sox10* as central regulator of oligodendroglial differentiation had been inactivated using the CRISPR/Cas9 system[25]. QrtPCR analysis revealed that *Nfatc2* is strongly downregulated upon loss of endogenous Sox10 (Fig. 3e). Recovery of *Nfatc2* after restoration of Sox10 expression by lentiviral transduction confirms Sox10 dependence.

This dependence correlates with the presence of ChIP-Seq peaks (GEO accession number GSE64703) for Sox10 in the *Nfatc2* locus[26]. Sox10 was found to be enriched in the distal of the two promoters of the *Nfatc2* gene that is preferentially used in oligodendroglial cells[22], and in an evolutionary conserved region (ECR) 87 kb downstream of the transcriptional start site (ECR87) (Fig. 3g). Luciferase reporter assays confirmed that both the distal promoter as well as ECR87 were activated by Sox10 in transiently

transfected N2a cells (Fig. 3f). Bioinformatic analysis of the 0.7 kb ECR87 revealed the presence of 9 potential Sox10 binding sites. Two of them were confirmed in EMSA (Fig. 3h and Suppl. Fig. 3e, f). Mutational inactivation of binding sites 2 and 3 (Suppl. Fig. 3e, f) strongly decreased Sox10-responsiveness of ECR87 in luciferase assays performed in transiently transfected N2a cells (Fig. 3i). Retroviral transduction of a tdTomato reporter under ECR87 control in cortical slices of newborn mice confirmed the ability of ECR87 to direct reporter gene expression to oligodendroglial cells in situ with more than 75% of tdTomato-expressing cells being Sox10-positive (Fig. 3j). Control retrovirus failed to direct reporter gene expression to Sox10-positive cells (see Fig. 4g). Together these data define ECR87 as a Sox10-dependent oligodendroglial enhancer and Nfatc2 as a direct target of Sox10 in oligodendroglial cells.

**Nfatc2 does not influence myelin gene expression directly.** Sox10 induces several stage-specific transcription factors in myelinating glia that then cooperate with Sox10 in positive feed-forward loops to permit terminal differentiation and myelination[11, 27–29]. As our studies pointed to a role of Nfat proteins in oligodendroglial differentiation, we first performed luciferase reporter gene assays in transiently transfected N2a cells to study the impact of Nfatc2 by itself or in the presence of Sox10 on several regulatory regions of myelin genes (Suppl. Fig. 4a). For these experiments we used a constitutively active non-phosphorylatable Nfatc2 version. None of the tested myelin regulatory elements, including enhancers of the *Mbp*, *Plp1*, and *Mog* genes, and the promoters of the *Gjb1/connexin-32* and *Gjc2/connexin-47* genes was activated by Nfatc2. Additionally, Nfatc2 did not enhance Sox10-dependent stimulation of these regulatory regions. Instead, Sox10-dependent activation rates remained unchanged or were even reduced. This argues against a direct influence of Nfatc2 on myelin gene expression and is in line with our earlier observation that Nfat signaling primarily influences the early stages of differentiation.

**Expression of *Nkx2.2* is regulated by Sox10-responsive ECRs.** The Nkx2.2 transcription factor is induced during these early phases and later required for differentiation[8]. Accordingly, *Nkx2.2* transcript levels increased in cultured mouse oligodendroglial cells during the first 24 h of differentiation (Fig. 4a). However, this increase is substantially blunted in the presence of 1 µM VIVIT. This and the reduction of Nkx2.2-positive cells in oligodendrocyte-specific *CnB1* knockout mice (Fig. 2i and Suppl. Fig. 2i) suggests that Nfat proteins affect Nkx2.2 expression in differentiating oligodendrocytes.

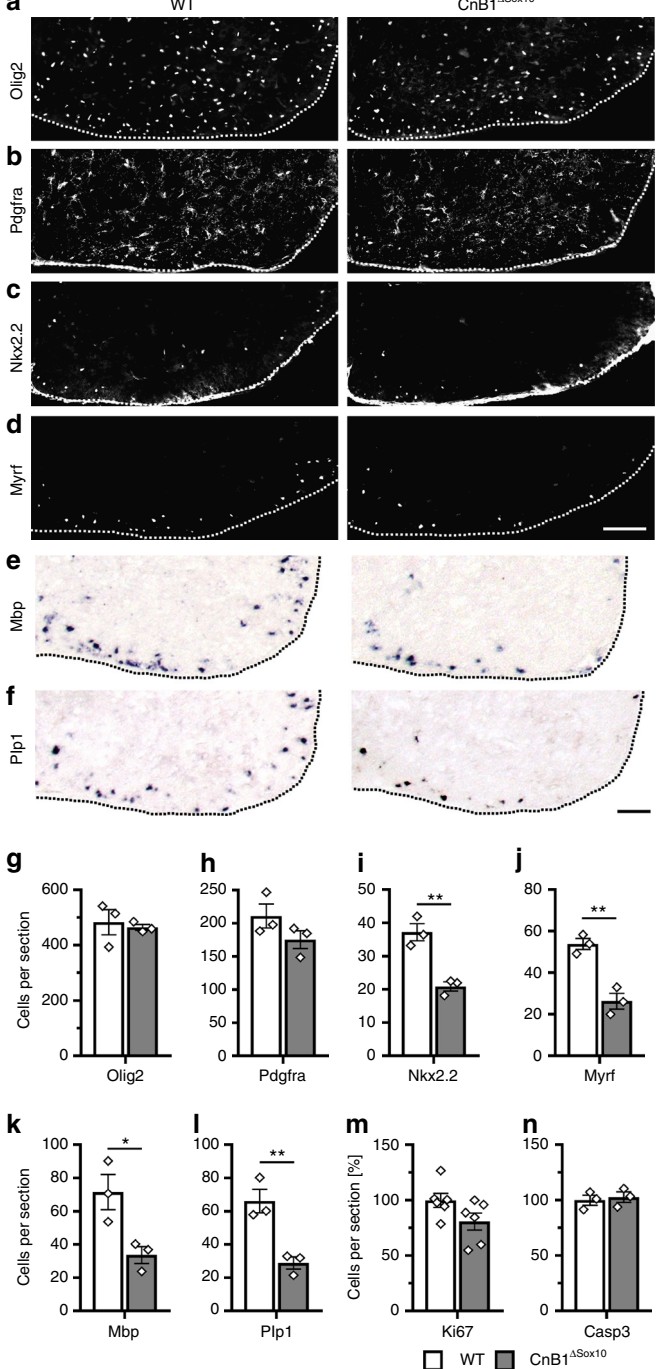

**Fig. 2** Nfat/calcineurin signaling is required for oligodendroglial differentiation in vivo. **a–f** Analysis of oligodendroglial marker expression in 18.5 dpc-old embryos of wildtype (WT) and CnB1$^{\Delta Sox10}$ mice by immunohistochemistry and in situ hybridization. Antibodies were directed against Olig2 (**a**), Pdgfra (**b**), Nkx2.2 (**c**), and Myrf (**d**). Riboprobes recognized *Mbp* (**e**) and *Plp1* (**f**) mRNA. Scale bars, 100 µm. **g–n** From these and similar stainings for Ki67 (**m**) and cleaved caspase 3 (**n**) quantifications were performed on three or six embryos for each genotype ($n = 3$ for **g–l** and **n**, $n = 6$ for **m**) counting three separate sections. Presentations are as absolute (**g–l**) or relative (**m**, **n**) numbers (values: Olig2: 483 ± 46 for WT and 465 ± 11 for CnB1$^{\Delta Sox10}$; Pdgfra: 211 ± 18 for WT and 175 ± 13 for CnB1$^{\Delta Sox10}$; Nkx2.2: 37 ± 3 for WT and 21 ± 1 for CnB1$^{\Delta Sox10}$; Myrf: 54 ± 3 for WT and 26 ± 4 for CnB1$^{\Delta Sox10}$; Mbp: 72 ± 11 for WT and 34 ± 5 for CnB1$^{\Delta Sox10}$; Plp1: 66 ± 7 for WT and 29 ± 4 for CnB1$^{\Delta Sox10}$; Ki67: 100 ± 6.4 for WT and 80.8 ± 7.7 for CnB1$^{\Delta Sox10}$; cleaved caspase 3: 100 ± 4.5 for WT and 102.7 ± 4.8 for CnB1$^{\Delta Sox10}$). Statistical significance was determined by two-tailed Student's $t$-test (*$P \leq 0.05$; **$P \leq 0.01$; ***$P \leq 0.001$)

To test whether the *Nkx2.2* gene is a direct target of Nfat activity, we first tried to identify the active regulatory regions in differentiating oligodendrocytes. For this purpose, we consulted existing ChIP-Seq data sets for Sox10 and Olig2 in differentiating rat oligodendrocytes (GEO accession numbers GSE64703 and GSE42447) and localized binding sites for these transcription factors in a 300 kb interval surrounding the *Nkx2.2* gene (from −150 to +150 kb relative to the transcriptional start site)[26, 30]. We detected four ECRs with peaks for Sox10 and/or Olig2 (Fig. 4b). Among 14 genes that code for oligodendroglial transcription factors and underwent the same analysis, only *Nkx6.2*, *Myrf*, *Tcf7l2*, and *Sip1* exhibited a similar number of peaks (Suppl. Fig. 4b). Multiple peaks for both Sox10 and Olig2 were only found for *Nkx2.2* and the related *Nkx6.2* suggesting that the detection of multiple Sox10-binding and Olig2-binding ECRs near the *Nkx2.2* gene is significant. Eighty four percent (i.e., 26 out of 31) of the detected peaks were also positive for acetylated H3K27 indicating that they correspond to active enhancers.

Before studying the impact of Nfatc2 on the *Nkx2.2* ECRs, we investigated their response to Sox10 as a central regulator of oligodendroglial differentiation. When inserted into a luciferase reporter construct, all four ECRs from the mouse *Nkx2.2* gene conveyed concentration-dependent Sox10 responsiveness to a minimal promoter in transiently transfected N2a cells. However, Sox10-dependent activation was much more pronounced for the ECRs at −19 and +5 kb (ECR19 and ECR5) than for the ECRs at −115 and +45 kb (ECR115 and ECR45) (Fig. 4c). Therefore, we concentrated on these two enhancers.

ChIP-Seq data had not shown a Sox10 peak at ECR19 (Fig. 4b). However, our own ChIP experiments in cultured primary rat

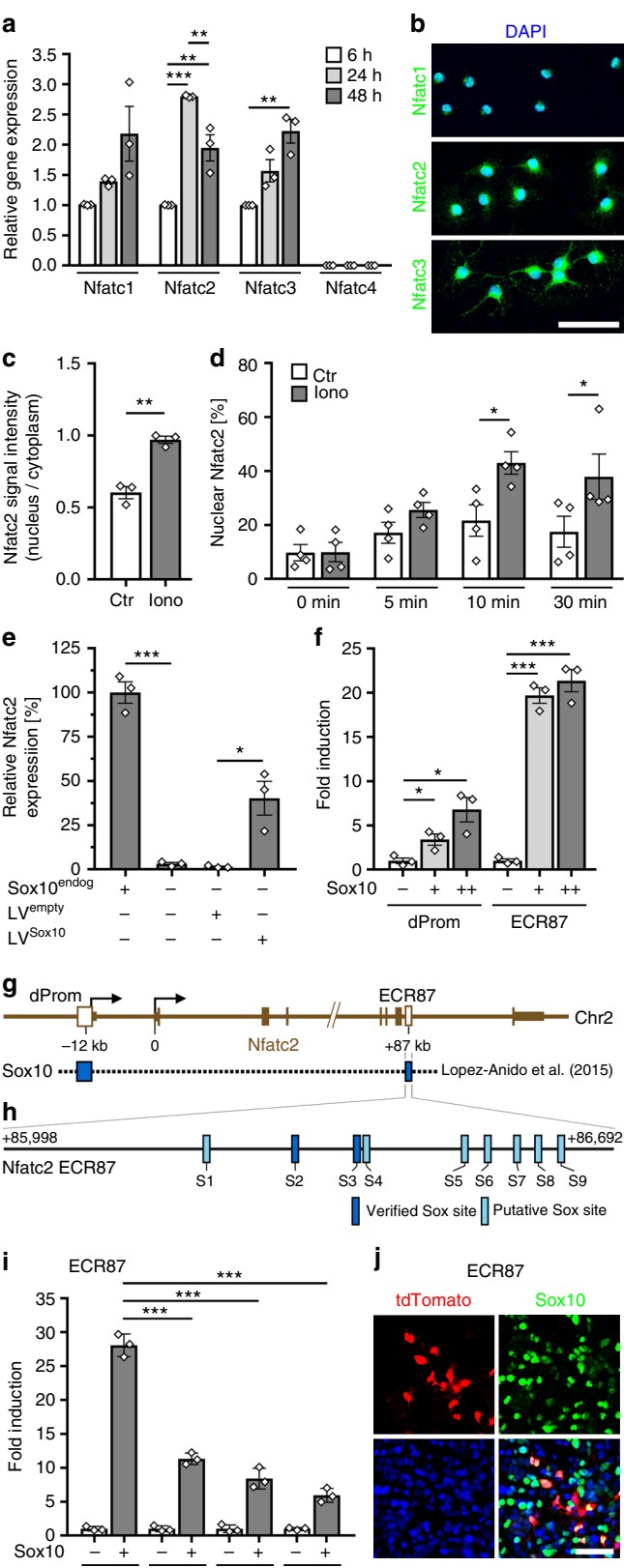

**Fig. 3** Nfats are regulated by Sox10 in oligodendroglia. **a** QrtPCR-determined *Rplp0*-normalized *Nfat* expression in mouse oligodendrocytes after 6 (open bars, set to 1), 24 (light grey bars) or 48 h (dark grey bars) differentiation ($n = 3$) (*Nfatc1*: 24 h 1.4 ± 0.1, 48 h 2.2 ± 0.5; *Nfatc2*: 24 h 2.8 ± 0.1, 48 h 2.0 ± 0.2; *Nfatc3*: 24 h 1.6 ± 0.2, 48 h 2.2 ± 0.2; *Nfatc4*: not detected). **b** Immunocytochemistry on differentiating DAPI-counterstained (blue) mouse oligodendrocytes after 48 h with anti-Nfat antibodies (green). **c** Nuclear versus cytoplasmic Nfatc2 signal intensity in rat oligodendroglia treated with ionomycin (grey bars, iono) or DMSO (open bars, Ctr) ($n = 3$, 30 cells analyzed per experiment) (values: 0.60 ± 0.07 for Ctr, 0.97 ± 0.04 for iono). **d** Nuclear fraction of Nfatc2-eYFP in transfected mouse oligodendroglial cells before and after treatment with ionomycin or DMSO ($n = 4$ with ~40 cells analysed per time point and condition) (values before and after 5, 10, and 30 min treatment: 9.7 ± 3.0%, 17.1 ± 3.9%, 21.6 ± 5.8%, and 17.4 ± 5.8% for Ctr; 9.9 ± 3.6%, 25.5 ± 2.8%, 43.0 ± 4.2%, and 37.9 ± 8.4% for ionomycin). **e** *Gapdh*-normalized *Nfatc2* quantification by qrtPCR in wildtype (Sox10endog +, set to 100%) or Sox10-deficient variant (Sox10endog −) Oln93 cells, before and after transduction with a control (LVempty) or Sox10-expressing (LVSox10) lentivirus ($n = 3$) (wildtype: 100.0 ± 6.1; variant: 3.1 ± 0.8; LVempty-infected variant: 1.5 ± 0.4; LVSox10-infected variant: 40.3 ± 9.6). **f** Sox10-dependent (+, light grey bars = low amounts; ++, dark grey bars = high amounts) fold inductions ± SEM of luciferase reporters carrying *Nfatc2* distal promoter (dProm) or ECR87 in transfected N2a cells after 48 h ($n = 3$) (dProm: 3.4 ± 0.6 and 6.8 ± 1.4; ECR87: 19.7 ± 0.9 and 21.4 ± 1.3). **g** Mouse *Nfatc2* locus with exons (brown boxes), transcription start sites (arrows), and Sox10 ChIP-Seq peaks (blue boxes). **h** Putative (light blue) and EMSA-confirmed (dark blue) Sox10 binding sites in ECR87. **i** Sox10-dependent (+, grey bars) fold inductions ± SEM of wildtype or mutant ECR87 luciferase reporters with inactivated Sox10 binding sites (S2m, S3m, S2/3m) in transfected N2a cells after 48 h ($n = 3$) (WT: 28.1 ± 1.0; S2m: 11.3 ± 0.5; S3m 8.4 ± 0.9; S2/3m 5.9 ± 0.6). **j** Sox10 immunohistochemistry (green) on cortical slices 7 days after transduction with a retroviral ECR87-tdTomato reporter (red). DAPI counterstain in blue. Scale bars, 50 μm. Statistical significance was determined by Bonferroni-corrected one-way (**a**) or two-way ANOVA (**d**) and two-tailed Student's *t*-test (**c**, **e**, **f**, **i**) (*$P \leq 0.05$; **$P \leq 0.01$; ***$P \leq 0.001$)

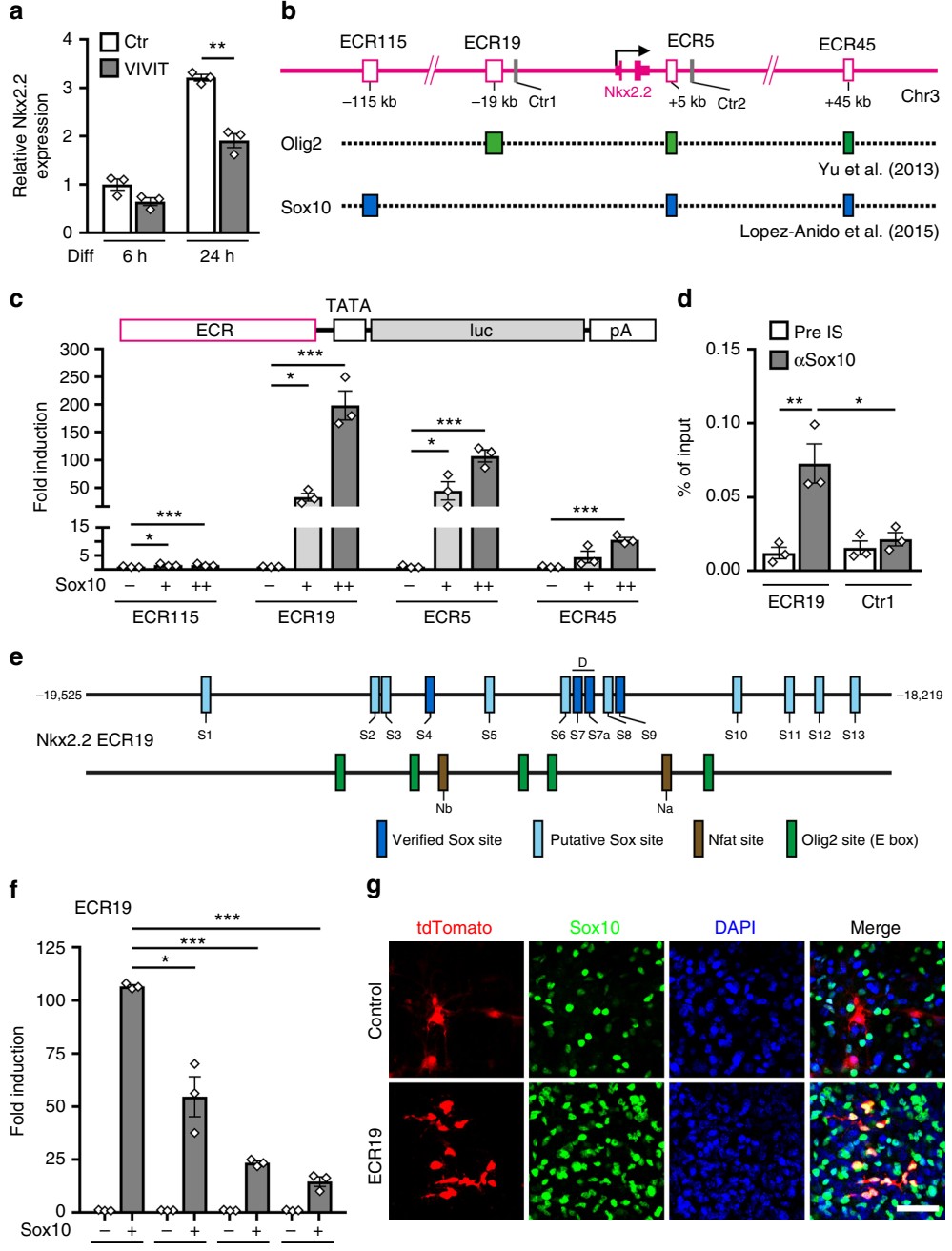

**Fig. 4** Sox10 activates oligodendroglial *Nkx2.2* expression via ECR19. **a** QrtPCR analysis of *Nkx2.2* expression in mouse oligodendroglial cultures after 6 and 24 h differentiation in absence (Ctr, open bars) or presence of 1 μM VIVIT (grey bars) (*n* = 3). *Nkx2.2* amounts under control conditions after 6 h were set to 1 (values: 1 ± 0.20 for Ctr and 0.65 ± 0.14 for VIVIT after 6 h, 3.21 ± 0.12 for Ctr and 1.92 ± 0.25 for VIVIT after 24 h). **b** Rat *Nkx2.2* locus with exons (solid pink boxes), ChIP-Seq peaks (GEO accession numbers GSE64703 and GSE42447) for Olig2 (green boxes) and Sox10 (blue boxes) in ECRs (open pink boxes) at −115, −19, +5, and +45 kilobases relative to the transcriptional start site (arrow) and ChIP control regions (Ctr1 and Ctr2, grey boxes). **c** N2a cell transfections with luciferase reporters carrying *Nkx2.2* ECRs in absence (−) or presence of low (+, light grey bars) and high Sox10 amounts (++, dark grey bars) (*n* = 3). Sox10-dependent fold inductions ± SEM were determined after 48 h with activities in the absence of Sox10 set to 1 (ECR115: 1.6 ± 0.3 at both concentrations; ECR19: 32.9 ± 7.1 and 198.3 ± 26.0; ECR5: 44.6 ± 16.4 and 107.3 ± 10.7; ECR45: 4.5 ± 2.1 and 10.6 ± 0.9). **d** ChIP on differentiating rat oligodendrocytes after 4 days (*n* = 3) using rabbit pre-immune (pre IS, open bars) and anti-Sox10 (αSox10, grey bars) antiserum. Amounts of immunoprecipitated ECR19 and control region (Ctr1) were qPCR-determined and are presented as percent of input (ECR19: 0.012 ± 0.004 for pre IS and 0.073 ± 0.013 for αSox10; Ctr1: 0.016 ± 0.005 for pre IS and 0.021 ± 0.005 for αSox10). **e** Localization of binding sites for Sox10 (putative, light blue; EMSA-confirmed, dark blue), Nfat (brown) and Olig2 (green) in ECR19. Numbers on left and right correspond to mouse ECR19 positions relative to transcriptional start site. For EMSA and sequences, see Suppl. Fig. 4c, d. **f** N2a cell transfections with wildtype (WT) or mutant ECR19 luciferase reporters with inactivated Sox10 binding sites (S4m, S7/7am, and S4/7/7am) in absence (−) or presence (+, grey bars) of Sox10 (*n* = 3). Sox10-dependent fold inductions ± SEM were determined 48 h post transfection (WT: 106.7 ± 0.7; S4m: 54.6 ± 9.1; S7/7am 23.4 ± 0.9; S4/7/7am 14.6 ± 2.3). **g** Sox10 immunohistochemistry (green) on cortical slices 7 days after transduction with retroviruses carrying tdTomato reporters (red) under control of a minimal promoter (control) or minimal promoter and ECR19 (ECR19). Co-expression is in yellow, DAPI counterstain in blue. Scale bar, 50 μm. Statistical significance was determined by two-tailed Student's t-test (*P ≤ 0.05; **P ≤ 0.01; ***P ≤ 0.001)

oligodendroglial cells found Sox10 selectively enriched at ECR19 relative to a control region in the vicinity indicating that Sox10 binding to ECR19 may have simply been missed in the earlier ChIP-Seq studies (Fig. 4d). Bioinformatic analysis identified 13 potential binding sites for Sox10 monomers or dimers in the 1.3 kb ECR19 (Fig. 4e). Three of these sites bound Sox10 in electrophoretic mobility shift assays (EMSA) (Fig. 4e and Suppl. Fig. 4c, d). Sites 4 and 9 bound a Sox10 monomer, site 7/7a a Sox10 dimer. Site-directed mutagenesis of the strongly binding site 4 and site 7/7a to a sequence without Sox10-binding capacity (Suppl. Fig. 4c, d) substantially reduced the ability of ECR19 to stimulate reporter gene expression in a Sox10-dependent manner in transiently transfected N2a cells (Fig. 4f). Retroviral transduction of a tdTomato reporter under ECR19 control in cortical slices of newborn mice confirmed the ability of ECR19 to direct reporter gene expression to oligodendroglial cells in situ with more than 90% of the tdTomato-expressing cells being Sox10-positive (Fig. 4g). In the absence of ECR19, only few cells expressed the control tdTomato reporter and more than 80% were negative for Sox10. This argues that ECR19 is not only Sox10-dependent, but also a bona fide oligodendroglial enhancer in rodents.

Analogous experiments were also carried out for ECR5. ChIP confirmed selective Sox10-binding to ECR5 in cultured primary rat oligodendroglial cells as compared to an adjacent control region (Suppl. Fig. 5a). Of the bioinformatically identified 14 potential binding sites in the 0.8 kb ECR5, four (i.e., site 3, site 7, site 8a, and site 9) were bound by Sox10 monomers in EMSA (Suppl. Fig. 5b–d). Mutational inactivation of site 3, site 8a, and site 9 each reduced the ability of ECR5 to impart Sox10-dependent activation on a luciferase reporter in transiently transfected N2a cells (Suppl. Fig. 5e). Mutation of site 8a had by far the biggest impact, followed by mutation of site 9 and site 3. These results strongly suggest that ECR5 also functions as a Sox10-dependent oligodendroglial enhancer in rodents. In fact, all our data indicate that ECR5 behaves very similar to ECR19. For space saving reasons we concentrate in the following on the ECR19 results.

**Nfatc2 assists in Sox10-stimulated Nkx2.2 expression**. To study the impact of Nfat proteins and calcineurin signaling on the expression of Nkx2.2, we analyzed the effect of Nfatc2 on the *Nkx2.2* ECR19 enhancer in transiently transfected N2a cells (Fig. 5a). A constitutively active non-phosphorylatable Nfatc2 version did not influence expression of a luciferase reporter under control of ECR19. However, it increased the Sox10-dependent activation from 58-fold to 130-fold indicating that Nfatc2 amplifies the Sox10 effect by cooperating with this transcription factor.

Bioinformatic analysis revealed the presence of potential recognition motifs for Nfat proteins in ECR19 (Fig. 4e). To study the cooperation mechanistically we performed EMSA with the two predicted Nfat binding sites and confirmed binding (Suppl. Fig. 6a, b). ChIP provided additional support for Nfat binding, as ECR19, but not a neighbouring control fragment was enriched in chromatin immunoprecipitated from cultured oligodendroglial cells with two anti-Nfatc2 antibodies relative to the pre-immune or IgG control (Suppl. Fig. 6c). Mutational inactivation of either of the identified sites strongly reduced the ability of Nfatc2 to augment Sox10-dependent activation of the ECR19 enhancer in transiently transfected N2a cells (Suppl. Fig. 6d, e). Inactivation of both sites abolished synergism completely arguing that Nfatc2 has to physically bind to ECR19. Reduced Sox10-dependent activation in the absence of

Nfat sites and co-transfected Nfatc2 points to cooperative interactions of Sox10 with endogenous Nfat proteins in N2a cells.

To confirm in vivo relevance of this finding, we electroporated the neural tube of Hamburger-Hamilton stage (HH) 11 chicken embryos with expression plasmids for Sox10 and constitutively active Nfatc2, and checked for induction of endogenous Nkx2.2 two days later in GFP-labeled electroporated cells (Fig. 5b, c). In agreement with published results[12], Sox10 was able to induce Nkx2.2 in approx. 20% of electroporated cells. Similarly, Nfatc2 induced Nkx2.2 in approx. 25% of the electroporated cells. When both Sox10 and Nfatc2 were electroporated, the rate of Nkx2.2 induction was significantly increased to 39% without alterations in the absolute number of electroporated cells (Fig. 5b, c). Considering further that $89.5 \pm 1.3\%$ ($P \leq 0.001$, $n = 3$) of all electroporated cells were double positive for Sox10 and Nfatc2, the increase is likely a consequence of joint activation by Sox10 and Nfatc2. We ascribe the additive rather than synergistic character of the activation to the high protein levels that accumulate in electroporated cells.

**Nkx2.2 and Olig2 repress each other**. According to ChIP-Seq data, the Sox10-dependent *Nkx2.2* enhancers also bind Olig2 (Fig. 4b). Bioinformatic analyses confirmed the presence of potential Olig2-binding sites (Fig. 4e). Therefore, we asked how the presence of Olig2 affects Sox10 activity on these enhancers. In transiently transfected N2a cells, Olig2 substantially reduced the ability of Sox10 to stimulate ECR19-dependent reporter gene expression (Fig. 5d). A similar effect was also observed for ECR5. Olig2 thus counteracts the Sox10-dependent stimulation of the Nkx2.2 enhancers.

We had previously identified OLEa as the core of a Sox10-dependent enhancer that mediates *Olig2* expression in oligodendroglial cells[10]. Using this enhancer, we studied the influence of Nkx2.2 on *Olig2* expression. Intriguingly, Nkx2.2 interfered with stimulation of an OLEa-containing reporter construct by Sox10 in transiently transfected N2a cells (Fig. 5e), just as Olig2 interfered with the Sox10-dependent stimulation of an ECR19-containing reporter. This argues that factors in addition to Sox10 are needed to relieve cross-repression of Olig2 and Nkx2.2.

To confirm in vivo relevance of cross-repression in oligodendroglial cells, we made use of a mouse model in which Sox10 expression is induced at moderate levels throughout the CNS by combining a *Brn4::Cre* transgene, a *Rosa26*$^{stopflox-tTA}$ allele, and a *TetSox10* transgene (Fig. 6a)[10]. In these mice, *Brn4::Cre*-dependent activation of the tetracycline-controlled transactivator (tTA) in neuroepithelial ventricular zone cells leads to permanent tTA-dependent Sox10 expression in these cells and their progeny. In line with Sox10 being an activator of Olig2 and Nkx2.2, we observed induction of these transcription factors in cells that ectopically express Sox10. At 12.5 days post coitum (dpc), Olig2 and Nkx2.2-expressing cells were predominantly localized in the dorsal half of the spinal cord (Fig. 6b). However, these cells represented only a small fraction of the cells with ectopic Sox10 expression. They usually expressed either Olig2 or Nkx2.2 and not both. Low rate of induction and absence of co-expression support the assumption that Olig2 and Nkx2.2 repress each other and that Sox10 alone is not sufficient to overcome cross-repression in vivo.

**Nfat proteins relieve cross-repression of Olig2 and Nkx2.2**. As Nfat proteins are active in differentiating oligodendrocytes where Olig2 and Nkx2.2 occur, we wondered how Nfat proteins would act in their presence. For some of the experiments we used constitutively active Nfatc2, for others constitutively active calcineurin CnA/Ppp3ca that works by activating endogenous Nfat

proteins. As expected from experiments with constitutively active Nfatc2 (Fig. 5a), active calcineurin did not influence *Nkx2.2* ECR19 activity in transiently transfected N2a cells, but augmented Sox10-dependent stimulation of this enhancer (Fig. 6c). Intriguingly, this enhancement was sufficient to compensate for the negative impact of Olig2 on the Sox10-dependent ECR19 stimulation and to restore full ECR19 activation in the

presence of Olig2. Thus, Nfat proteins are capable of relieving the inhibitory effects of Olig2 on Sox10-dependent ECR19 activation.

We also investigated the impact of Nfat proteins and active calcineurin on the *Olig2* OLEa enhancer. As already observed for the *Nkx2.2* ECR19 enhancer, constitutively active Nfatc2 alone was unable to stimulate reporter gene expression via OLEa in transiently transfected N2a cells (Fig. 6d). Nfatc2 also did not alter Sox10-dependent activation of OLEa. Similarly, activated

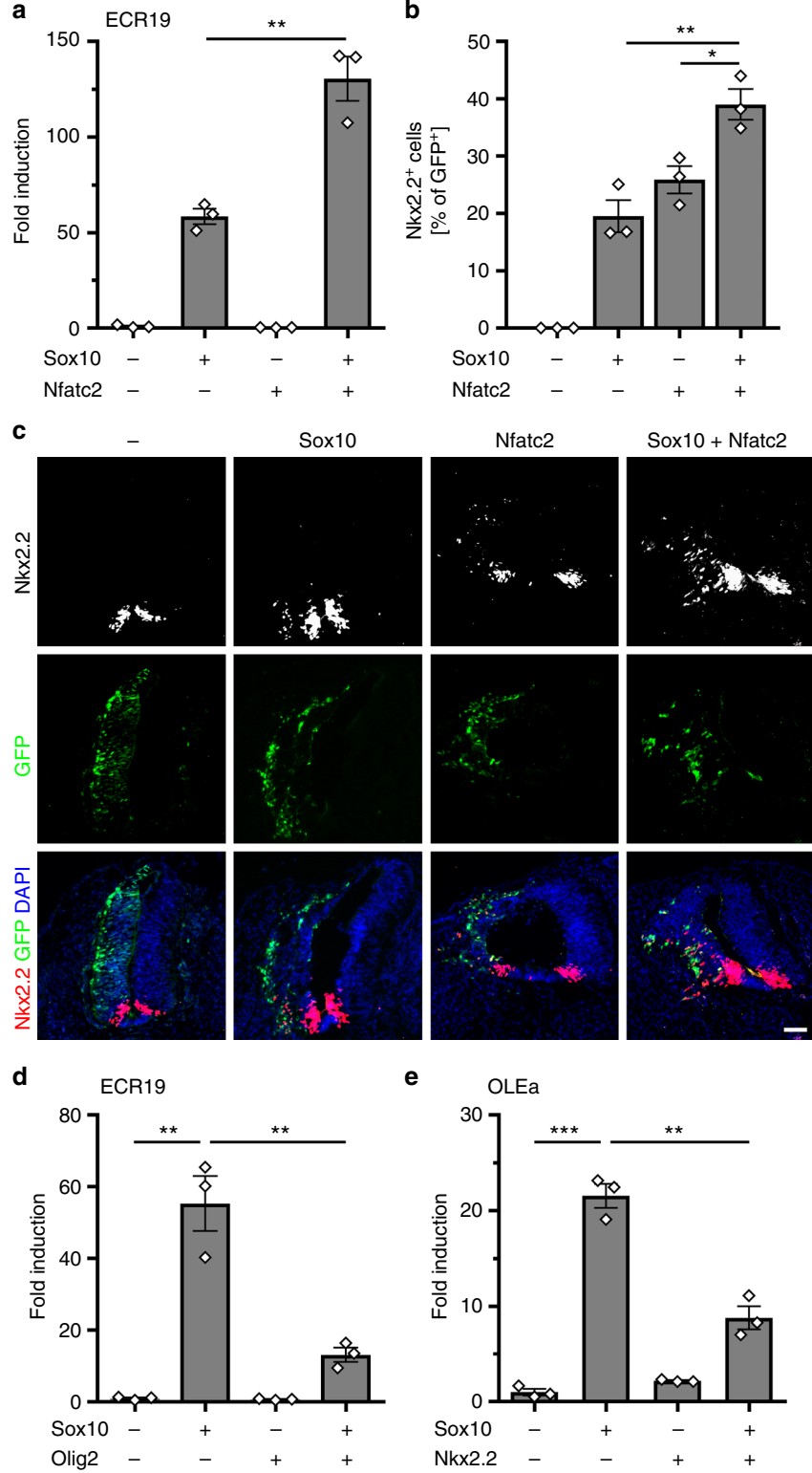

calcineurin signaling did not influence OLEa activity by itself or in the presence of Sox10 (Fig. 6e). However, it was able to counteract the repressive effects of Nkx2.2 on the Sox10-dependent OLEa activation (Fig. 6e). This demonstrates that Nfat proteins are also instrumental in relieving the repressive effects of Nkx2.2.

As further control, we analyzed protein levels in N2a cells transfected with various transcription factor combinations (Suppl. Fig. 6f, g). Protein levels varied substantially among experiments because of different transfection efficiencies. Taking this caveat into account, Western blotting did not yield any evidence that Olig2 or Nkx2.2 reduced Sox10 levels in transiently transfected cells as a means of lowering Sox10 activity. We also did not find any evidence that Nfat proteins exert their effects on enhancer activity by increasing Sox10 levels. Moreover, there was no evidence from retroviral transduction of cultured rat oligodendroglial cells that overexpression of Nkx2.2 or Olig2 negatively impacts endogenous Sox10 levels (Suppl. Fig. 6h).

If Nfat proteins help to relieve cross-repression of Nkx2.2 and Olig2, we would expect that co-expression of constitutively active Nfatc2 with Sox10 in the chicken neural tube not only increases the number of cells that ectopically express Nkx2.2 (Fig. 5b, c), but also the number of cells that express both Nkx2.2 and Olig2. Therefore, we re-assessed our chicken electroporation experiments. The number of double positive cells was indeed significantly increased after co-electroporation of Sox10 and Nfatc2 (Fig. 6f).

Assuming that this effect of Nfat proteins on expression of Nkx2.2 and Olig2 is relevant for their role in oligodendroglial differentiation, we would furthermore predict that ectopic expression of Nkx2.2 in differentiating oligodendrocytes is capable of rescuing the consequences of calcineurin inhibition on differentiation and myelin gene expression. Therefore, we retrovirally transduced Olig2-positive OPCs, differentiated them for 6 days in the absence or presence of the calcineurin inhibitor FK506 and determined the percentage of transduced cells with Mbp expression (Fig. 6g). When oligodendroglial cells were not exposed to FK506, rates of Mbp-expressing cells were high and not dependent on forced Nkx2.2 expression. Treatment with FK506 led to a substantial reduction of Mbp expression in oligodendroglial cells transduced with control retrovirus. Nkx2.2 overexpression efficiently rescued this reduction (Fig. 6g).

**NFAT activity is relevant in human oligodendrocytes**. To analyze the role of calcineurin signaling and NFAT proteins in humans, induced pluripotent stem cell (iPSC)-derived oligodendrocytes (iOL) were generated[31]. After 28 days of differentiation, transcripts were detected for all four members of the NFAT family (Fig. 7a). Amounts increased substantially for *NFATc2* relative to neural precursor cells (NPCs) and remained fairly stable for all other family members. By immunocytochemistry, NFATc1, NFATc2, NFATc3 as well as NFATc4 were detected in nuclei of MBP-positive human iOL (Fig. 7b).

To determine whether NFATs are involved in the differentiation of human oligodendrocytes, we differentiated iOL in the presence of VIVIT for variable time periods (Fig. 7c–g). Titration experiments revealed that VIVIT did not affect iOL viability at concentrations below 5 μM (Suppl. Fig. 7a). Accordingly, continuous exposure of iOL to 0.5 μM VIVIT for 21 days left cell numbers unaffected (Suppl. Fig. 7b). The same treatment, however, resulted in a significant 30–40% reduction of O4-positive human iOL (Fig. 7c, e, Suppl. Fig. 7c, d). MBP-positive human iOL were comparably reduced after 35 days of treatment with 0.5 μM VIVIT (Fig. 7d, f). Addition of VIVIT during days 21 to 35 of differentiation only led to a mild reduction in the number of MBP-positive iOL that had been generated from O4-positive iOL, sorted by FACS at day 21 (Fig. 7g). Similar effects on O4 and MBP expression were obtained when iOL were differentiated in the presence of 10 μM FK506 (Fig. 7h–j), again without detectable effects on iOL viability and cell numbers (Suppl. Fig. 7e, f). These results argue that differentiation of human oligodendrocytes also depends on calcineurin-dependent NFAT activity, particularly in the initial stages of differentiation.

Finally, we analyzed NFAT expression in human CNS biopsies. NFAT proteins were detected in white matter samples from MS patients and from individuals without inflammatory demyelinating disease in several cell types (Fig. 7k–m and Suppl. Fig. 7g–n). To determine whether NFAT family members are also expressed in oligodendroglial cells, we performed double immunohistochemistry for OLIG2 as an oligodendroglial lineage marker and NOGOA as a marker for mature oligodendrocytes. Of the four NFAT proteins, NFATc3 and NFATc4 were unequivocally detected in nuclei of NOGOA-positive mature oligodendrocytes (Fig. 7k–m). NFATc3 was additionally present in GFAP-positive astrocytes, whereas NFATc4 was found in neurons and CD68-positive microglia/blood derived monocytes (Suppl. Fig. 7l–n). NFATc2 was predominantly detected in microglia/blood derived monocytes (Suppl. Fig. 7k). When MS tissue samples were analyzed, we observed a significantly reduced number of NOGOA-positive oligodendrocytes expressing NFATc4 in active demyelinating and post-demyelinating lesions compared to control and periplaque white matter where the vast majority of oligodendrocytes expressed NFATc4. The number of NFATc4-positive oligodendrocytes furthermore appeared to recover during remyelination (Fig. 7n).

**Fig. 5** Sox10-dependent enhancer activation is stimulated by Nfatc2 and inhibited by Olig2 and Nkx2.2. **a** Transient transfections of N2a cells with luciferase reporters carrying *Nkx2.2* ECR19 in the absence (−) or presence (+) of Sox10 and constitutively active Nfatc2 as effectors (*n* = 3). Activation of reporter gene expression was determined in extracts 48 h post transfection and is presented as fold inductions ± SEM with transfections in the absence of effectors arbitrarily set to 1 (58.5 ± 4.0 for Sox10, 0.3 ± 0.0 for Nfatc2, and 130.5 ± 11.6 for Sox10 in combination with Nfatc2). **b, c** Neural tube electroporations in HH11-stage chicken embryos (*n* = 3). The relative number of electroporated cells, in which Nkx2.2 expression was induced 48 h after neural tube electroporation of Sox10 and Nfatc2, was quantified (**b**: 0 for control, 19.5 ± 4.9 for Sox10, 25.9 ± 4.1 for Nfatc2, and 39.0 ± 4.6 for Sox10 in combination with Nfatc2). Transverse sections were used for quantifications (**c**). Sections were probed for the occurrence of Nkx2.2 (white in upper row, red in lower row). Electroporated cells on the left side all express GFP (green) and to variable extent Nkx2.2 (white in upper row, red in lower row). Nuclei are counterstained by DAPI (blue). Scale bar, 50 μm. **d, e** Transient transfections of N2a cells with luciferase reporters carrying ECR19 (**d**) or OLEa (**e**) in the absence of added transcription factors (−) or in the presence (+) of Sox10, Olig2, Nkx2.2, and combinations thereof (*n* = 3). Activation of reporter gene expression was determined in extracts 48 h post transfection and is presented as fold inductions ± SEM with transfections in the absence of added transcription factors being arbitrarily set to 1 (ECR19: 55.3 ± 7.7 in the presence of Sox10, 0.7 ± 0.1 in the presence of Olig2 and 13.1 ± 2.0 in the presence of Sox10 and Olig2; OLEa: 21.6 ± 1.3 in the presence of Sox10, 2.2 ± 0.1 in the presence of Nkx2.2 and 8.8 ± 1.2 in the presence of Sox10 and Nkx2.2). Statistical significance was determined by two-tailed Student's *t*-test (*$*P \leq 0.05$; **$P \leq 0.01$; ***$P \leq 0.001$).

**Discussion**

In this study we identify calcineurin signaling and its Nfat effectors as novel determinants of oligodendrocyte differentiation in vitro and in vivo (Fig. 7o). We used pharmacological inhibition in rodent cell culture and in organotypic slices to document the importance of calcineurin signaling, and confirmed our findings by mouse mutagenesis. We extended our observations from rodents to humans by analyzing human iPSC-derived oligodendrocytes, and established relevance for MS as the most frequent demyelinating disease in humans. Our results also point to differences in the occurrence and functional relevance of Nfat proteins between rodents and humans. Such species-specific differences in expression patterns are not uncommon. They have for instance been described recently on a global scale between mouse

and human astrocytes[32]. However, despite these minor differences, the role of calcineurin signaling in oligodendroglial differentiation is conserved.

We also provide insight into the molecular mode of action by showing that Nfat proteins are instrumental in resolving cross-repression of Olig2 and Nkx2.2 as Sox10 partners (Fig. 7o). Under conditions of active calcineurin signaling, Nfat proteins provide the necessary synergistic support to Sox10 for Nkx2.2 induction in the presence of Olig2. This is essential in the early phases of oligodendrocyte differentiation[5, 6, 8, 9].

It needs to be emphasized that induction of Nkx2.2 in differentiating oligodendrocytes bears the danger of a concomitant loss of Olig2 expression. Therefore it is important that Sox10 and Nfat proteins also overcome the Nkx2.2-dependent

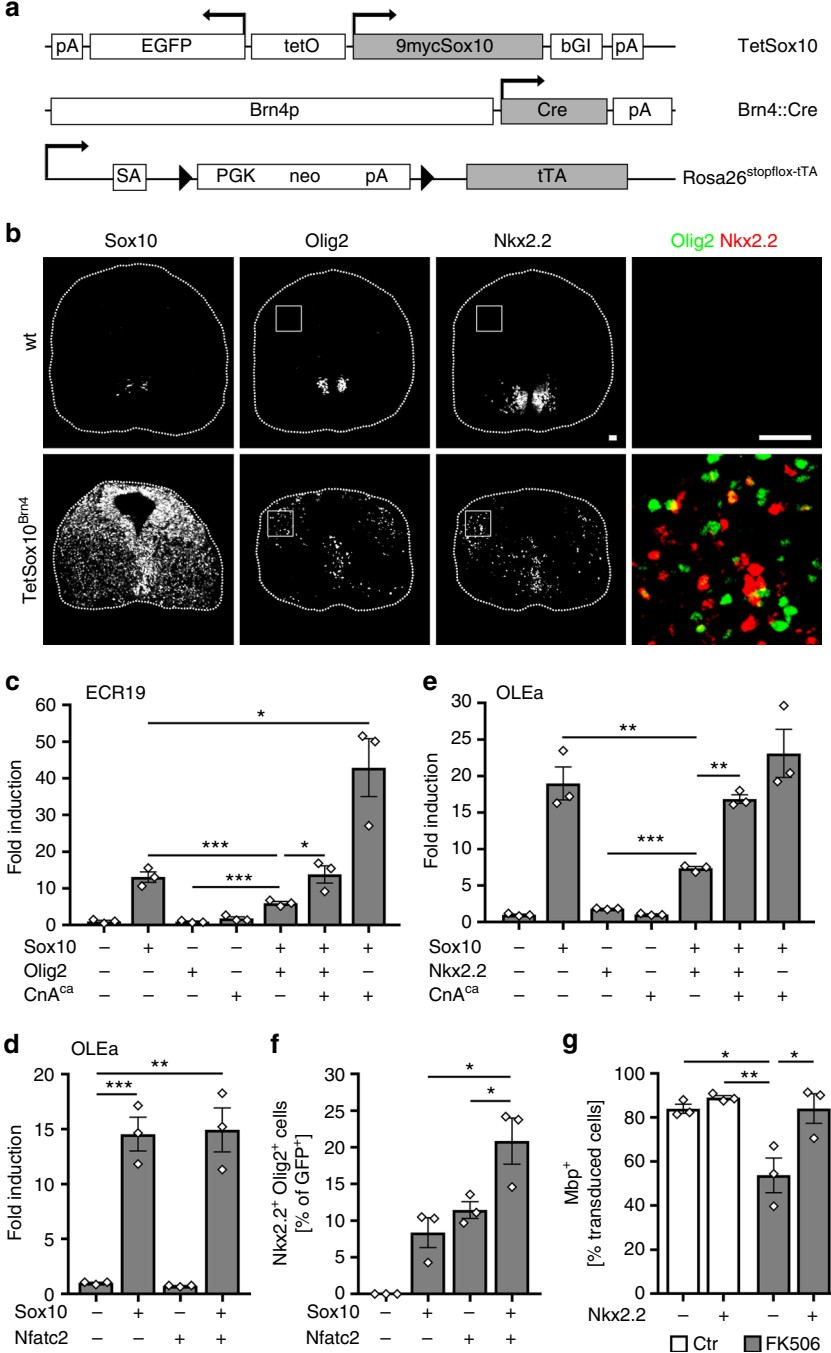

repressive effect on Olig2 expression as shown in our study. Nfat function in oligodendrocytes is thus very different from their proposed role in Schwann cell development where Olig2 and Nkx2.2 are not expressed and where Nfats have been suggested to function downstream of neuregulin signaling in myelin gene expression[33].

In our study we also elucidated the regulatory elements that mediate the joint effects of Sox10 and Nfat proteins. In case of *Nkx2.2*, we characterized ECR19 as an upstream enhancer that jointly binds Sox10 and Nfat proteins and mediates the synergistic effect. In case of *Olig2*, the effect is mediated through the previously identified OLEa enhancer[10]. A closer look at the complex interactions of Sox10 and Nfat proteins on both enhancers furthermore reveals that there are slight differences in Nfat action. In case of ECR19, Nfat proteins synergistically stimulate Sox10 activity and thereby compensate the repressive effect of Olig2. In case of OLEa, Nfat proteins do not synergistically stimulate Sox10 activity but rather counteract the repressive effects of Nkx2.2.

At least Nfatc2 as a highly expressed family member is under direct control of Sox10 in rodent oligodendroglial cells and this effect is mediated by ECR87, a downstream enhancer that probably cooperates with the distal of two alternative promoters in Sox10-dependent gene activation. Sox10 starts to be expressed in the oligodendroglial lineage immediately after specification[34] and already induces Nfat proteins in OPCs. However, data obtained in vitro by VIVIT treatment of oligodendroglial cultures and in mice following oligodendroglial Nfat inactivation argue that Nfat proteins become active in the early stages of differentiation. This goes along with an increased expression and an increased nuclear localization of Nfat proteins at this time, likely upon calcineurin activation.

Calcineurin is usually activated by increases in intracellular calcium levels[16]. Such increases occur at the onset of oligodendroglial differentiation, likely mediated by voltage-operated calcium or calcium-permeable glutamate channels[35, 36]. Calcineurin may integrate these and other signals by sensing the overall intracellular calcium concentrations and transforming them into Nfat activity. We find it particularly intriguing that Nfat proteins confer susceptibility to extracellular signals to the oligodendroglial regulatory network (Fig. 7o), and thus provide a means of linking extrinsic and intrinsic regulation of oligodendroglial differentiation and myelination.

Importantly, we provide the first evidence that Nfat proteins are not only involved in the differentiation of rodent oligodendrocytes, but also of human oligodendrocytes. This immediately raises the question about relevance for disease, in particular for patients suffering from demyelinating conditions, such as MS. In inflammatory and demyelinating MS lesions we observed a significant downregulation of NFATc4 in oligodendrocytes. This may be explained as a reaction to an inflammatory milieu in which oligodendroglial cells downregulate cellular processes that are not absolutely required for survival. Our in vitro results and mouse studies predict that such a downregulation of NFATc4 will also affect oligodendroglial differentiation and remyelination. In fact, impaired remyelination is frequently observed in MS and may be at least in part caused by altered NFAT activity[1–3].

Nfats are also known to be pivotal in the activation of inflammatory cells, especially T lymphocytes. In experimental autoimmune encephalomyelitis as an animal model of MS, deficiency of Nfatc1 and/or Nfatc2 ameliorated demyelination suggesting that blocking NFAT activity might be a treatment strategy in MS[37]. However, our results suggest that such an approach might be detrimental for oligodendroglial differentiation and remyelination. This assumption is supported by observations of leukoencephalopathy as frequent side-effect in transplanted patients treated with FK506 or cyclosporine A to suppress the immune system and prevent rejection of transplanted organs[38–40]. We like to propose that demyelination in FK506-treated patients is at least in part caused by interference with calcineurin and NFAT activity in differentiating oligodendrocytes.

In summary, identification of Nfat proteins as important contributors to oligodendroglial differentiation substantially broadens our understanding of the underlying regulatory network and points to disturbed NFAT activity as cause of immunologically or pharmacologically induced demyelination or failed remyelination.

## Methods

**Plasmids and viruses**. Expression plasmids for Sox10, Olig2, Nkx2.2, Nfatc2 (in wildtype and constitutively active version) and constitutively active CnA/Ppp3ca were based on pCMV5 and/or pCAGGS-IRES-nlsGFP. Most have been described before[7, 34, 41]. Those for Nfatc2 contained coding sequences derived from Addgene plasmids # 11791 and # 11792 (gift from Anjana Rao, La Jolla). The expression plasmid for Nfatc2-eYFP and H2B-mCherry were kindly provided by A. Flügel and D. Lodygin[23]. Coding sequences for Sox10 were additionally inserted into pEF1α-IRES-GFP to produce Sox10-expressing lentiviruses. Coding sequences for Nkx2.2

**Fig. 6** Nfat proteins overcome cross-repression of Olig2 and Nkx2.2 in cooperation with Sox10. **a** *TetSox10*, *Brn4::Cre* and *Rosa26^{stopflox-tTA}* alleles for CNS overexpression of Sox10. Arrows mark transcription start sites, triangles loxP sites. 9mycSox10, myc-tagged Sox10 coding sequences; bGI β-globin intron, Brn4p *Brn4* promoter, Cre Cre coding sequences, EGFP enhanced GFP coding sequences, neo neomycin resistance cassette, pA polyadenylation site, PGK phosphoglycerate kinase promoter, SA splice acceptor, tetO bidirectional tetracycline-responsive promoter, tTA tetracycline-controlled transactivator coding sequences. **b** Immunohistochemistry for Sox10, Olig2, and Nkx2.2 on transverse spinal cord sections of wildtype (wt) and Sox10 overexpressing (*TetSox10^{Brn4}*) embryos at 12.5 dpc. Right panels are magnifications of boxed areas with Olig2 in green and Nkx2.2 in red. Scale bars, 50 μm. **c** N2a cell transfections with ECR19 luciferase reporter in absence (−) or presence (+) of Sox10, constitutively active CnA (CnA^{ca}) and Olig2 (*n* = 3). Fold inductions ± SEM were determined after 48 h with reporter activity in the absence of effectors set to 1 (values: 13.1 ± 1.4 for Sox10, 0.9 ± 0.2 for Olig2, 1.8 ± 0.5 for CnA^{ca}, 6.0 ± 0.5 for Sox10 and Olig2, 13.9 ± 2.4 for Sox10, Olig2 and CnA^{ca}, and 42.9 ± 7.9 for Sox10 and CnA^{ca}). **d** N2a cell transfections with *Olig2* OLEa luciferase reporter in absence (−) or presence (+) of Sox10 and Nfatc2 (*n* = 3) (14.6 ± 1.7 for Sox10, 0.7 ± 0.0 for Nfatc2 and 14.9 ± 2.0 for Sox10 and Nfatc2) or (**e**) in absence (−) or presence (+) of Sox10, CnA^{ca} and Nkx2.2 (*n* = 3) (values: 19.0 ± 2.3 for Sox10, 1.8 ± 0.1 for Nkx2.2, 1.0 ± 0.1 for CnA^{ca}, 7.3 ± 0.3 for Sox10 and Nkx2.2, 16.9 ± 0.6 for Sox10, Nkx2.2 and CnA^{ca}, and 23.1 ± 3.3 for Sox10 and CnA^{ca}). **f** Neural tube electroporations in HH11-stage chicken embryos (*n* = 3) and quantification of the relative number of electroporated cells, in which joint expression of Nkx2.2 and Olig2 was induced after 48 h (values: 0 for control, 8.4 ± 3.5 for Sox10, 11.5 ± 2.0 for Nfatc2, and 20.9 ± 5.4 for Sox10 and Nfatc2). **g** Immunohistochemical determination of the percentage of Mbp-expressing rat oligodendroglia transduced with GFP-expressing control (−) or Nkx2.2-overexpressing retrovirus (+), after 6 days of differentiation in absence (Ctr, open bars) or presence of 1 μM FK506 (grey bars) (*n* = 3 separate cultures). (Values: 84.0 ± 3.6% in GFP-transduced untreated cultures; 89.0 ± 1.7% in Nkx2.2-transduced untreated cultures; 53.7 ± 13.7% in GFP-transduced FK506-treated cultures; 84.1 ± 11.7% in Nkx2.2-transduced FK506-treated cultures). Statistical significance was determined by two-tailed Student's *t*-test (**c–f**) and Bonferroni-corrected one-way ANOVA (**g**) (*$P \leq 0.05$; **$P \leq 0.01$; ***$P \leq 0.001$)

and Olig2 were inserted into pCAG-IRES-GFP[42] to produce Nkx2.2-expressing or Olig2-expressing retroviruses.

ECRs from the *Nkx2.2* and *Nfatc2* genes had the following positions in Mm10: ECR115 chr2:147,308,082-147,306,872, ECR19 chr2:147,205,927-147,204,621, ECR5 chr2:147,181,491-147,180,670, ECR45 chr2:147,133,155-147,132,254, and ECR87 chr2:168,504,367-168,503,673. All ECRs were obtained by PCR from mouse ES cell DNA and inserted into the pGL2 luciferase reporter plasmid in front of a β-globin minimal promoter. Luciferase reporter plasmids with regulatory regions

from myelin genes were as described[11, 43] and included Mm10 sequences chr18:82,538,010-82,538,640 (*Mbp*), chrX:136,826,509-136,827,680 (*Plp1*), chr17:37,018,818-37,018,178 (*Mog*), chrX:101,383,449-101,383,864 (*Gjb1*) or chr11:59,183,437-59,182,710 (*Gjc2*).

ECR19 and ECR87 were additionally placed in front of β-globin minimal promoter and tdTomato sequences. The resulting expression cassette was used to replace the CAG-IRES-GFP sequences in pCAG-IRES-GFP to produce retroviruses. Another cassette used to replace CAG-IRES-GFP sequences for

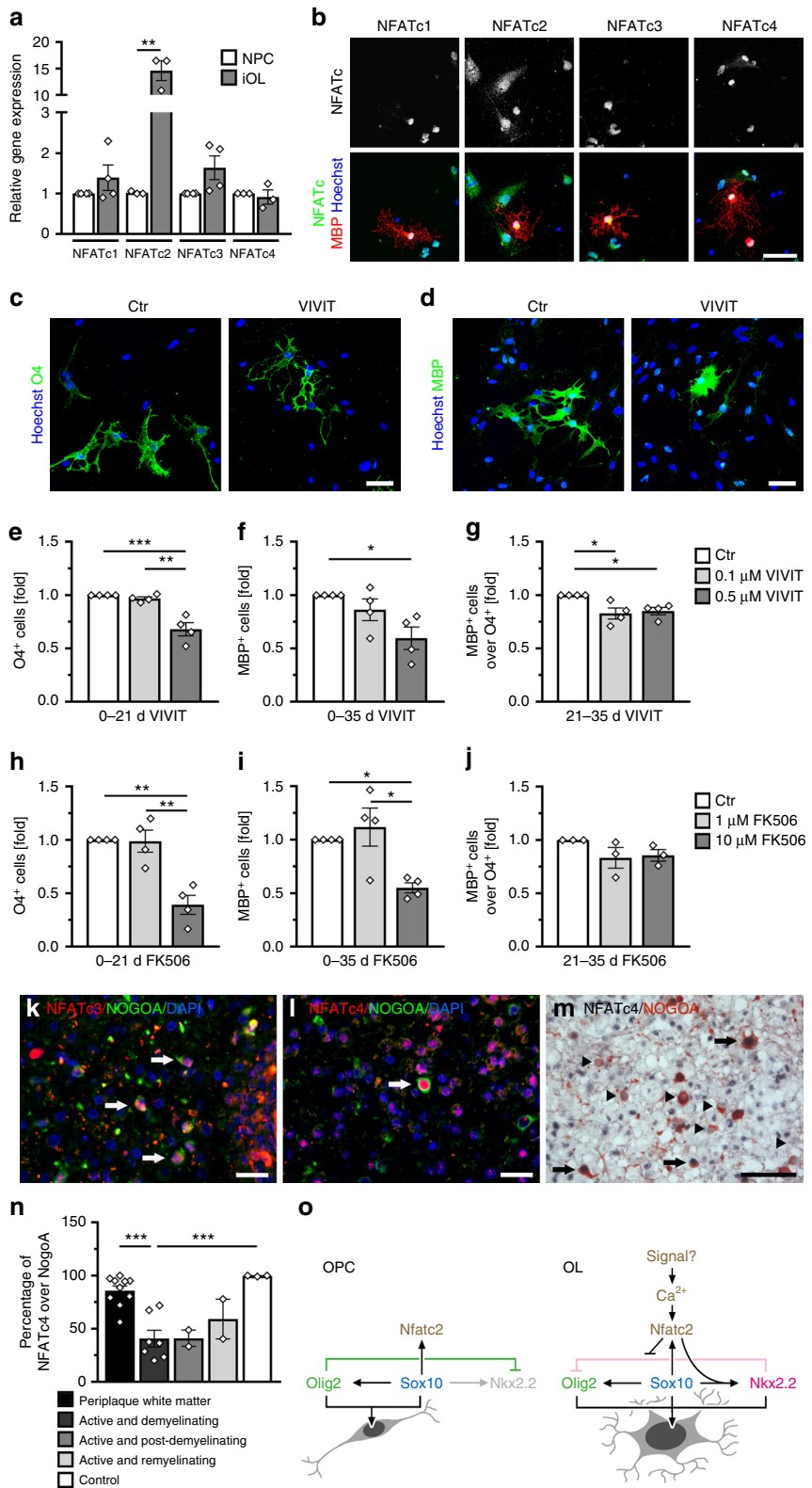

retrovirus production included two copies of the SV40 polydenylation signal followed by four copies of the Nfat responsive element from the human *IL2* gene in front of β-globin minimal promoter and tdTomato sequences. The distal promoter of the *Nfatc2* gene (positions chr2:168,602,300-168,601,567 in Mm10) was inserted immediately in front of the luciferase reporter in pGL2. The OLEa-containing luciferase reporter has been described before[10]. Site-directed mutagenesis was performed by PCR (Q5® Site-Directed Mutagenesis Kit, New England Biolabs) to substitute Sox and Nfat binding sites in ECRs (see Suppl. Figs. 3d, 4d and 5d) by unrelated recognition sites for restriction enzymes.

**Cell culture assays**. Both primary cells and cell lines were used. Cell lines included human embryonic kidney 293 cells (obtained from ATCC), mouse N2a neuroblastoma (obtained from ATCC), rat Oln93 oligodendroglial cells (gift of C. Richter-Landsberg, Oldenburg), and mouse Oli-neu cells (gift of J. Trotter, Mainz). Only Oln93 and Oli-neu cells were checked for mycoplasma contamination. N2a, Oln93, and Oli-neu cells were authenticated by PCR to establish their neural and oligodendroglial identity, respectively. The 293 cells were not authenticated. They were only used for extract preparation after transfection with polyethylenimine[34] or virus production after transfection with Lipofectamine 2000 (Thermo Fischer Scientific). Most cell lines were grown in DMEM supplemented with 10% fetal calf serum (FCS). Oli-neu cells were kept in SATO medium containing 1% horse serum. N2a cells were transfected with Superfect reagent (Qiagen) and used for luciferase assays[10]. Oln93 cells were employed in wildtype version or as a clone that had undergone CRISPR/Cas9-dependent *Sox10* inactivation[25]. Reintroduction of Sox10 in this clone was conducted by lentiviral transduction at a MOI of 1 followed by fluorescence-activated cell sorting (FACS) of transduced GFP-expressing cells on a MoFlo cell sorter (Beckman Coulter). For Nfat translocation studies, Oli-neu cells were transfected using FuGENE® HD (Promega). Their differentiation was induced 24 h after transfection by 1 mM dibutyryl-cAMP.

Rodent primary oligodendroglial cells were obtained from brain tissue of 6–9 day-old male C57BL/6 mice or newborn Wistar rats of both sexes. Mouse primary OPCs were purified by sequential immunopanning using plates coated with anti-BSL 1 Griffonia simpliconia lectin (Vector Labs) followed by plates coated with anti-CD140a (Biolegend)[44]. They were maintained under proliferating conditions on poly-L-lysine in defined serum-free medium supplemented with 10 ng/ml PDGF-AA and 5 ng/ml NT3 (Peprotech). Differentiation was induced by replacing PDGF-AA by 10 ng/ml CNTF. Mouse OPCs were transfected with Viafect™ Transfection Reagent (Promega). Rat OPC cultures were prepared from mixed glial cultures by shake-off[45]. For proliferation, rat OPCs were grown on poly-ornithine in defined medium containing 10 ng/ml PDGF-AA and 10 ng/ml Fgf2. Rat OPCs were transduced at a MOI of 0.5–1. For differentiation, growth factors were replaced by 40 ng/ml T3 and 0.5% FCS.

For subcellular localization studies of Nfatc2, rat and mouse OPCs were used. Rat OPCs were treated for 1 h with 1 μM ionomycin or DMSO as the ionomycin solvent and stained with antibodies directed against Nfatc2 and Sox10 before determination of cytoplasmic and nuclear Nfatc2 signals on confocal images using NIH imageJ software. Mouse OPCs were transfected with an expression plasmid for Nfatc2-eYFP and H2B-mCherry, and one day after transfection treated for 5–30 min with 0.5 μM ionomycin or DMSO. Fluorescent signals for Nfatc2-eYFP and H2B-mCherry were evaluated in single cells on photographs by Adobe Photoshop CS5 software and the fraction of Nfatc2-eYFP signal that overlapped with the nuclear mCherry signal was determined. The same methods of analysis were also applied to compare subcellular Nfatc2 localization in OPCs and differentiating oligodendrocytes.

Human iPSC-derived oligodendrocytes were generated as described[31]. Briefly, iPSCs (described and characterized in ref[46]) were converted into NPCs via embryoid bodies[47]. Both iPSCs and NPCs were checked for mycoplasma contamination. NPCs were then cultured on matrigel (Corning) and infected with a polycistronic lentiviral vector containing the coding regions of human SOX10, OLIG2, NKX6.2 followed by an IRES-pac cassette allowing puromycin selection. After one day of recovery, medium was changed to a glial induction medium containing B27 and N2 supplements, SAG, NT3, PDGF-AA, FGF2, IGF-I, and 15 ng/ml T3. This day was defined as day 0 of differentiation. On the third day puromycin selection started for 5 days. At day 4 of differentiation glial induction medium was replaced by glial differentiation medium containing B27 and N2 supplements, NT3, IGF-I, 60 ng/ml T3, and 50 μM dibutyryl-cAMP. Medium was changed every other day.

Where indicated, rodent or human oligodendroglial cultures were incubated with 0.1–5 μM of the NFAT inhibitor 11R-VIVIT (Merck Millipore), 0.1–10 μM of the calcineurin inhibitor FK506 (Bertin Pharma), 0.5–1 μM ionomycin (Cell Signaling) or 0.05–0.1% DMSO as solvent control over different time periods.

Cell viability was measured according to manufacturer's protocol using CellTiter-Glo® Luminescent Cell Viability Assay (Promega). Proliferation was determined using BrdU incorporation (Cell Proliferation ELISA, BrdU [colorimetric], Roche Diagnostics).

**Fluorescence-activated cell sorting**. For FACS of human iOLs, anti-O4-APC was used according to manufacturer's instruction (Miltenyi). After accutase treatment, cells were collected and scattered. Following separation with a 40 μm cell strainer, cells were counted and incubated with anti-O4-APC at 4 °C for 10 min. After washing cells underwent FACS on a FACSAria IIu cell sorter (BD Biosciences) and were immediately seeded in glial differentiation medium. O4-positive cells were identified by using unstained cells and isotype controls. For quantification, DAPI was added to exclude dead cells and at least $1 \times 10^5$ cells were quantified.

**Immunocytochemistry**. Immunocytochemistry was performed essentially as described and involved cell fixation in 4% paraformaldehyde, permeabilization (except for O4 staining), blocking, and consecutive incubation with primary and secondary antibodies, separated by extensive washing cycles[31, 48]. Nuclei were counterstained with 4′,6-diamidin-2-phenylindole (DAPI) or Hoechst dye. The following primary antibodies were used: mouse anti-O4 monoclonal (R&D systems, #MAB1326, Lot# HWW1115081, 1:100 dilution), mouse anti-Nfatc1 monoclonal (Santa Cruz, clone H-10, #sc-17834, Lot# D0110, 1:25 dilution), rabbit anti-Nfatc2 antisera (Sigma, #HPA008789, Lot# A33084, 1:50 dilution, and Immunoglobe #0112-02, Lot# 358, 1:50 dilution), rabbit anti-Nfatc3 antiserum (Santa Cruz, clone M-75, #sc-8321, Lot# L2612, 1:50 dilution), rabbit anti-Nfatc4 antiserum (Santa Cruz, clone L-9, #sc-32985, Lot# H0306, 1:50 dilution), rat anti-MBP monoclonal (Abcam, #ab7349, Lot# GR188102-12, 1:50 dilution and Bio-Rad, #MCA409S, Lot# 210610, 1:750 dilution). Secondary antibodies were coupled to Cy3 (Dianova, 1:500 dilution) or Alexa-Fluor (Molecular Probes, 1:1000 dilution) fluorescent dyes. Stainings were by default documented on a Leica DMI6000 B inverted microscope or an Olympus IX50 inverted microscope. For generation of confocal images, a Zeiss LSM 700 or a Leica TCS SL confocal microscope was used. At least 100 cells were analyzed for each condition and every independent experiment. If not stated otherwise, positive cells were determined as percentage of DAPI/Hoechst-positive cells.

**RNA isolation and qrtPCR**. For qrtPCR, total RNA was prepared, reverse transcribed and subjected to PCR on a CFX96 Real Time PCR System (Bio-Rad) or a

**Fig. 7** NFATs are relevant in human oligodendrocytes. **a** *NFAT* expression in human iOLs (grey bars) and NPCs (open bars) by qrtPCR after normalization to *GAPDH*. Average *NFAT* levels in NPCs were set to 1 (*NFATc1*: 1.4 ± 0.3; *NFATc2*: 14.6 ± 1.9; *NFATc3*: 1.6 ± 0.3 and *NFATc4*: 0.9 ± 0.2). **b** Immunocytochemistry of MBP-positive (red) iOL with antibodies directed against NFATs (white in upper row, green in lower row); counterstaining with Hoechst dye (blue). **c–g** Analysis of differentiating iOL cultured 21 (**c**, **e**, **h**) or 35 days (**d**, **f**, **g**, **i**, **j**) in absence (Ctr, open bars) or presence of 0.1 (light grey bars) to 0.5 μM (dark grey bars) VIVIT (**c–g**, *n* = 4) or 1 (light grey bars) to 10 μM (dark grey bars) FK506 (**h–j**, *n* = 3-4). VIVIT/FK506 incubation was continuous (**c–f**, **h**, **i**) or from days 21–35 (**g**, **j**). Cultures were stained with anti-O4 (green, **c**) and anti-MBP antibodies (green, **d**). From staining and FACS analysis (see Suppl. Fig. 7c, d) the fraction of O4-positive cells after 21 days (**e**, **h**) and MBP-positive cells after 35 days (**f**, **i**) was determined. For treatment during days 21–35, the fraction of FACS-sorted O4-positive cells was determined that had reached a MBP-positive stage (**g**, **j**). The relative number of marker-positive cells under control conditions was set to 1 and used to normalize in pairwise fashion (values: 1 for control, 0.97 ± 0.01 for 21 days 0.1 μM VIVIT, 0.68 ± 0.06 for 21 days 0.5 μM VIVIT, 0.86 ± 0.10 for 35 days 0.1 μM VIVIT, 0.60 ± 0.10 for 35 days 0.5 μM VIVIT, 0.82 ± 0.05 for 0.1 μM VIVIT from days 21–35, 0.86 ± 0.04 for 0.5 μM VIVIT from days 21–35, 0.99 ± 0.21 for 21 days 1 μM FK506, 0.39 ± 0.17 for 21 days 10 μM FK506, 1.12 ± 0.36 for 35 days 1 μM FK506, 0.55 ± 0.09 for 35 days 10 μM FK506, 0.83 ± 0.17 for 1 μM FK506 from days 21–35 and 0.86 ± 0.10 for 10 μM FK506 from days 21–35). **k–n** NFAT immunohistochemistry in human brain tissue. Antibodies directed against NFATc3 and NFATc4 (red in **k** and **l**, black in **m**) were combined with anti-NOGOA antibodies (green in **k** and **l**, red in **m**). Arrows mark double positive cells, arrowheads NOGOA-positive oligodendrocytes without NFATc4. **n** Percentage of NOGOA-positive oligodendrocytes with nuclear NFATc4 in periplaque white matter (86 ± 4%), active and demyelinating lesions (41 ± 8%), active and post-demyelinating lesions (41 ± 8%), active and remyelinating plaques (59 ± 19%), and control CNS tissue (99 ± 1%). **o** Model of oligodendroglial Nfat action. Statistical significance was determined by two-tailed Student's *t*-test (**a**) or Bonferroni-corrected one-way ANOVA (**e–j**, **n**) (*$P \leq 0.05$; **$P \leq 0.01$; ***$P \leq 0.001$). Scale bars, 50 μm

StepOne Plus real time cycler (Applied Biosystems). The melting curve of each sample was measured to ensure specificity of the amplified products. All samples were processed as technical triplicates. Data were analyzed by the ΔΔCt method[49]. The following primer pairs were used: 5′-TGGGAAAACAGTGACAACCA-3′ and 5′-CGCGTGTTCTTTCTGCCTAT-3′ for rat Nfatc2, 5′-CCTTCGGAAGGGT GCCTTTT-3′ and 5′-CGGCTGCCTTCCGTCTCATAG-3′ for mouse Nfatc1, 5′-G TTTCGGAGCTTCAGGATGC-3′ and 5′-CTACATGGAGAACAAGCCT-3′ for mouse Nfatc2, 5′-CTACTGGTGGCCATCCTGTTGT-3′ and 5′-ACTTTTGTG CTGGCGATTAT-3′ for mouse Nfatc3, 5′-CATTGGCACTGCAGATGAG-3′ and 5′-CGTAGCTCAATGTCTGAAT-3′ for mouse Nfatc4, 5′-GTACAAGGACTCAC ACACGAG-3′ and 5′-GTTCGAGGTGTCACAATGTTCT-3′ for mouse Mbp, 5′-ACCGCCTTCAACCTGTCTGT-3′ and 5′-CTCGTTCACAGTCACGTTGC-3′ for mouse Mag, 5′-CAAGACCTCTGCCAGTATAG-3′ and 5′-AGATCAGAACTT GGTGCCTC-3′ for mouse Plp1, 5′-CGACATGCAGGACACAATCA-3′ and 5′-G AGTAGCCGCTGGTTATGCT-3′ for mouse neurofilament (Nefl), 5′ ACCAA-CACAAAGACAGGGTT 3′ and 5′ CCGTGCAGGGAGTATTGGAG 3′ for mouse Nkx2.2, 5′-CCAAGGTCATTTTCGTGGAG-3′ and 5′-GGTCAGTTTTCGCTTC-CATC-3′ for human NFATc1, 5′-TATTACCTGCGGGGGTGAC-3′ and 5′-CCAG CTAAGGTGTGTGTCTATCA-3′ for human NFATc2, 5′-ACTCGTCTTTGGCGA GGA-3′ and 5′-TCATCTGGCTCAAGATCTGC-3′ for human NFATc3 and 5′-GG TATCACGCTGGAGGAAGT-3′ and 5′-CCAGGTGATGACAGTTCACG-3′ for human NFATc4. Depending on the species, transcript levels were normalized to rat Gapdh (amplified with 5′-TCCAGTATGACTCTACCCACG-3′ and 5′-CACGACA TACTCAGCACCAG-3′), mouse Rplp0 (amplified with 5′-CGACCTGGAAGTCC AACTAC-3′ and 5′- ATCTGCTGCATCTGCTTG-3′), and human GAPDH (amplified with 5′-CTGGTAAAGTGGATATTGTTGCCAT-3′ and 5′-TGGAAT-CATATTGGAACATGTA AACC-3′).

**Luciferase assays and Western blots.** For luciferase assays in N2a cells, 0.5 μg of pCMV5-based expression plasmid and 0.5 μg of luciferase reporter were used per 3.5 cm plate. Expression plasmids for single factors were 0.05 μg or 0.5 μg. Overall amounts of plasmid in a particular experiment were kept constant by adding empty pCMV5 where necessary. Whole cell extracts were prepared 48 h post transfection by lysing cells in 88 mM MES pH 7.8, 88 mM Tris pH 7.8, 12.5 mM MgOAc, 2.5 mM ATP, 1 mM DTT, and 0.1% Triton X-100. Luciferase activities were determined in the presence of 0.5 mM luciferin in 5 mM KHPO₄ pH 7.8 by detection of chemiluminescence. For Western blots, whole cell extracts were prepared by lysing cells in 10 mM HEPES pH 7.9, 10 mM KCl, 0.1 mM EDTA, 0.1 mM EGTA. After addition of NP-40 to 1% final concentration and NaCl to 400 mM, 15 min rotation at 4 °C and 5 min centrifugation, glycerol was added to a final concentration of 10% and whole cell extracts were used for Western blotting[41]. Images of Western blots have been cropped for presentation. Full size images are presented in Supplementary Fig. 8.

**Electrophoretic mobility shift assays.** Full length Sox10 and constitutively active Nfatc2 were produced in 293 cells transfected with 10 μg pCMV5-based expression plasmid per 100 mm plate. Extracts were prepared from transfected and mock-transfected 293 cells. With these extracts, EMSA were performed using ³²P-labeled 25 bp double-stranded oligonucleotides containing putative Sox and Nfat binding sites from ECR19 and ECR5 of the Nkx2.2 gene and ECR87 of the Nfatc2 gene (see Suppl. Figs. 3D, 4B and 5D)[34]. Oligonucleotides containing sequences of sites B and C/C′ from the Mpz gene served as controls for monomeric and dimeric binding of Sox10[50]. The positive control oligonucleotide for Nfat binding contained the Rag1 promoter consensus site (positions chr2:101,649,759-101,649,784 in Mm10). For Sox10 binding studies poly-dGdC was used as unspecific competitor, in case of Nfatc2 binding studies poly-dIdC.

**Chromatin immunoprecipitation.** Chromatin immunoprecipitation (ChIP) was performed essentially as described[11]. Chromatin was prepared from cultured rat oligodendroglial cells treated with 1% formaldehyde and sheared to fragments of approximately 300–600 bp in a Bioruptor (Diagenode). After quantification and pre-clearing, chromatin was incubated with rabbit antiserum against Sox10[51], corresponding pre-immune serum, polyclonal rabbit antiserum against Nfatc2 (Cell signaling technology, #5861S, Lot# 004), monoclonal rabbit anti-Nfatc2 antibody (ImmunoGlobe, 0112-02, Lot# 358) or control IgG before addition of protein A sepharose beads and precipitation. Crosslinks in precipitated chromatin were reversed and DNA was purified by proteinase K treatment, phenol/chloroform extraction, and ethanol precipitation. Detection and quantification of specific genomic regions in precipitated DNA was by qPCR on a Bio-Rad CFX96 thermocycler with each reaction performed in triplicate. The ΔΔCt method was used to calculate the percent recovery of a given DNA segment relative to the total input. Primers used to detect DNA sequences from the Nkx2.2 genomic locus included: 5′-GTTTCCATTCTGGGGGGAAAT-3′ and 5′-ACAAGTGCCTGTGG-GAAGTC-3′ to amplify positions −18,490 to −18,305 relative to the transcriptional start site (ECR19), 5′-CTGTCACTGGG TCCCTTTTGT-3′ and 5′-CTCCATCCTCCTCCCCAATA-3′ to amplify positions +5125 to +5313 (ECR5), 5′-ATGCAGGCAAAACATGTACG-3′ and 5′-ACAG TCACAGCACGTGTTCC-3′ to amplify positions −61,155 to −60,920 (Ctr), 5′-GC

ATATTTTGTAAGTGTCTCTGTGG-3′ and 5′-ATTGTCCCCCAGATGTTCAG-3′ to amplify positions −16,762 to −16,574 (Ctr1), and 5′-TTCTGACTTCGGTC CAACCT-3′ and 5′-GACCGAGGAGCGTCTCTAAA-3′ to amplify positions +7475 to +7644 (Ctr2).

**In ovo electroporation.** Fertilized chicken eggs were obtained from LSL Rhein-Main (Dieburg, Germany) and incubated in a humidified incubator at 37.8 °C. pCAGGS-IRES-nlsGFP–based expression plasmids were injected at a concentration of 2 mg/ml into the neural tube of chicken embryos at HH stage 11 before electroporation with five 50 ms pulses of 30 V using a BTX ECM830 electroporator. Transfected embryos were allowed to develop for 48 h before dissection[41].

**Transgenic mice.** To inactivate calcineurin signaling in the oligodendroglial lineage, floxed alleles for CnB1/Ppp3r1[17] were combined in mice with a Sox10::Cre Bac transgene[18, 19] or the Cnp1^Cre allele[20]. To overexpress Sox10 throughout the developing CNS, two copies of the TetSox10 transgene[10] were combined with the Rosa26^stopflox-tTA allele[52] and a Brn4::Cre[53]. Transgenic and wildtype mice were on a mixed C3H × C57Bl/6J background. They were kept under standard housing conditions with 12:12 h light-dark cycles and continuous access to food and water in accordance with animal welfare laws. Spinal cord tissue was obtained at 12.5 dpc and 18.5 dpc from male and female mouse embryos with relevant genotypes. Experiments were approved by the responsible local committees and government bodies (Regierung von Unterfranken and Landesamt für Natur, Umwelt und Verbraucherschutz Nordrhein-Westfalen).

**Cortical and cerebellar slice cultures.** For cortical slice culture, 300 μm thick coronal slices were generated from forebrains of 1–2 day old male C57Bl/6 mice at the level of the corpus callosum using a Leica VT1000S vibratome. Slices were placed on 0.4 μm Millicell-CM™ organotypic cell culture inserts (Merck-Millipore) and incubated for 7 days in medium containing 25% heat-inactivated horse serum (Gibco). Freshly prepared slices were either transduced by local retroviral injections into the lateral ventricles or treated with 1 μM FK506 or DMSO as solvent control. Slice culture medium was changed every second day, FK506 was freshly added on a daily basis.

For cerebellar slice culture, 300 μm thick sagittal slices of cerebellum and underlying hindbrain were generated from 1–3 day old male CD1 mice using a McIlwain tissue chopper. Slices were cultured for 12 days on Millipore Millicell-CM™ organotypic culture inserts and treated with 11-R VIVIT or 0.1% DMSO as solvent control. Medium was changed every 2–3 days.

**Human tissue samples.** Brain tissue samples were retrospectively investigated from 15 MS patients and three patients without pathology. Staining results in tissue samples from two MS patients were insufficient and therefore excluded from further analysis. None of the study authors was involved in decision-making with respect to biopsy. The sampled MS lesion areas were classified as active and demyelinating (n = 7), active and post-demyelinating (n = 2) or active and remyelinating (n = 2). For nine samples from MS patients' periplaque white matter was available[54]. The study was approved by the Ethics Committee of the University Hospital Münster.

**Immunohistochemistry and in situ hybridization.** Immunohistochemistry and in situ hybridization were performed on cultured slices and 10 μm cryotome sections of electroporated chicken neural tube or mouse spinal cord (forelimb level) after fixation in 4% paraformaldehyde, transfer to 30% sucrose and freezing in Tissue Freezing Medium (Leica) as described[11]. For in situ hybridization, DIG-labeled antisense riboprobes specific for Mbp and Plp1 were used[11]. Samples were analyzed and documented with a Leica MZFLIII stereomicroscope equipped with an Axiocam (Zeiss). For immunohistochemistry, the following primary antibodies were applied: rat anti-MBP monoclonal (Abcam, #ab7349, Lot# GR188102-12, 1:500 dilution and Bio-Rad, #MCA409S, Lot #210610, 1:500 dilution), mouse anti-Neurofilament L monoclonal (Dako, clone 2F11, Lot# 20014701, 1:500 dilution), mouse anti-Nkx2.2 monoclonal (Developmental Studies Hybridoma Bank, University of Iowa, clone 74.5A5, 1:5000 dilution), guinea pig anti-Sox10 antiserum (home made, validated on wildtype and knockout mouse tissue, 1:1000 dilution)[51], rabbit anti-Myrf antiserum (home made, validated on wildtype and knockout mouse tissue, 1:1000 dilution)[11], rabbit anti-Olig2 antiserum (Millipore, #AB9610, Lot# 2060464, 1:1000 dilution), rabbit anti-Pdgfra antiserum (Santa Cruz, #sc-338, Lot# E-1210, 1:300 dilution), rabbit anti-cleaved caspase 3 antiserum (Cell Signaling Technology, #9661, Lot# 0043, 1:200 dilution), rabbit anti-Ki67 antiserum (Thermo Fisher Scientific, #RM-9106, Lot# 9106S906D, 1:500 dilution), rabbit anti-GFP antiserum (Molecular Probes, #A11122, Lot# 1293114, 1:2000 dilution), goat anti-tdTomato antiserum (Sicgen, #AB8181-200, Lot# 8181240714, 1:1000 dilution). For anti-Nkx2.2 antibodies, signal intensity was enhanced by using the TSA-Plus Fluorescence system (PerkinElmer). Secondary antibodies were coupled to Cy3 (Dianova, 1:200 dilution) or Alexa488 (Molecular Probes, 1:500 dilution) fluorescent dyes. Nuclei were counterstained with DAPI. Immunohistochemical stainings were documented with a Leica DMI6000 B inverted microscope (Leica) equipped with a DFC 360FX camera (Leica). Confocal z-stacks of whole cerebellar

or cortical slices were acquired with a Zeiss LSM 700 or Zeiss LSM 780 confocal microscope, and images were analyzed using NIH ImageJ.

For human tissue, lesional, and non-lesional areas of paraformaldehyde-fixed, paraffin embedded human brain were sectioned. After antigen retrieval, slides were incubated overnight with mouse anti-NFATc1 monoclonal (Santa Cruz, clone H-10, #sc-17834, Lot# D0110, 1:50 dilution), rabbit anti-NFATc2 antiserum (Sigma, #HPA008789, Lot# A33084, 1:50 dilution), rabbit anti-NFATc3 antiserum (Sigma, #HPA023844, Lot# A83120, 1:50 dilution), rabbit anti-NFATc4 antiserum (Santa Cruz, clone L-9, #sc-32985, Lot# H0306, 1:50 dilution), rabbit anti-OLIG2 antiserum (IBL, #18953, Lot# 1B-327, 1:150 dilution), mouse anti-OLIG2 monoclonal (Medac # 387M-16, Lot# 1308812B, 1:100 dilution), mouse anti-NOGOA monoclonal (Clone 11c7, gift from M.E. Schwab, Brain Research Institute, ETH and University of Zürich, Switzerland, 1:10000 dilution), rabbit anti-NOGOA (Millipore, #AB5664P, Lot# 2903078, 1:200 dilution), mouse anti-GFAP monoclonal (eBioscience, clone GA5, #14-9892, Lot# E12755-103, 1:250 dilution) and mouse anti-CD68 monoclonal (Dako, #M0814, Lot #00095781, 1:100 dilution). For biotinylated secondary anti-rabbit (1:400 dilution, Vector) or anti-mouse (1:400 dilution; GE Healthcare) antibodies detection was by ExtrAvidin-Peroxidase (1:100 dilution; Sigma) using DAB as chromogen and hematoxylin as counterstain. Alternatively, Cy3-conjugated or Alexa488-conjugated antibodies were used in combination with a DAPI counterstain. Images were taken on an Olympus fluorescent microscope or a Zeiss LSM-710 confocal microscope with Zen software.

**Quantifications and statistical analysis**. Results from independent specimens or experiments were treated as biological replicates. Sample size was $n \geq 3$ for all molecular biology experiments and experiments using cell cultures, slice cultures or mice as common for this kind of study. For human brain tissue samples, sample size was $n = 2–9$ according to availability. No data were excluded from the analysis. Randomization was not possible. Investigators were blinded in cell culture experiments, not in animal experiments. GraphPad Prism6 (GraphPad software, La Jolla, CA, USA) was used to determine whether differences in cell numbers, luciferase activities, mRNA levels or immunoprecipitated DNA were statistically significant. These included two-tailed Student's $t$ tests or Bonferroni-corrected one-way or two-way ANOVA tests (*, $P \leq 0.05$; **, $P \leq 0.01$; ***, $P \leq 0.001$). The data met the assumptions of the chosen test. Variance between statistically compared groups was similar.

**Data availability**. All data generated or analyzed during this study are included in this published article and its supplementary information files.

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

## Acknowledgements
We thank Drs. E. Serfling (Würzburg), A. Flügel, and D. Lodygin (Göttingen) for reagents, Drs. J. Trotter (Mainz) and C Richter-Landsberg (Oldenburg) for cell lines, Drs. B. Crenshaw (Philadelphia), W.D. Richardson (London), and K.A. Nave (Göttingen) for mice. Beate Koch, Elke Hoffmann and Claudia Kemming are acknowledged for their technical expertise. Funded by the Bavarian State Ministry of Education and Culture, Science and Arts in the framework of the Bavarian Research Network Induced Pluripotent Stem Cells (ForIPS), the DFG (We1326/14, GRK2162; Ku 1477/6-1, SFB-TR128-B07) and the Interdisciplinary Clinical Research Center Münster (IZKF KuT3/012/15).

## Author contributions
T.K. and Mich.W. conceived and supervised the study. Ma.W., L.J.S., K.G., T.K. and Mich.W. designed the experiments. Ma.W., L.J.S., K.G. performed the experiments with the help of M.K., E.S., Mir.W., F.F., C.S., T.B., A.C.H., S.H., S.P., T.F., M.E., C.E., S.A., and A.J. M.S., J.W., and H.R.S. provided important information, reagents and materials. Ma.W., L.J.S., K.G., T.K., and Mich.W. analyzed the data. Ma.W., L.J.S., K.G., T.K., and Mich.W. wrote the manuscript.

## Additional information

**Competing interests:** The authors declare no competing interests.

