## [Peer Review File · Nature Communications]

Reviewers' comments:

Reviewer #1 (Remarks to the Author):

This is an interesting paper that describes a role for calcineurin signalling in oligodendrocyte differentiation by the activation of Nfat. It also shows that Nfat exerts its effects by altering the repressive interactions between the sox10, olig2, and nkx2.2 transcription factor network. The demonstration that this mechanism is also active in human cells, albeit with a smaller effect size, adds importance, as does the work showing a nfat expression in MS lesions. Overall the paper will be of significant interest to neuroscientists.

A problem, though, is the clarity. It is noteworthy that the abstract and start of the discussion described the work by starting with nfat and calcineurin, and then move on to the work on the enhancer binding sites. In contrast, the results describe the work in exactly the opposite way. Personally, I found the logic of the results hard to follow and think that the paper would be much easier to read and have more scientific impact if the results were ordered in the same way as the other sections of the paper.

If this is not done, then the logic for switching from nkx2 .2 enhancers on page 6 and the selection of nfat on page 7 ("caught our attention" is hardly scientific) needs much better explanation.

In terms of experiments these are well described, but I have four areas that require attention

First, how significant is it to find four peaks for transcription factor binding enhancers (page 5). What's the comparator here?

Second, the experiments in figure 4B and C don't demonstrate joint activation. The additive effect on nkx2.2 expression could simply reflect different cell populations being recruited.

Third, do the authors show that activated calcineurin increases nuclear localisation of nfatc2? This is an important experiment

Last, I suggest a figure summarising the pathway proposed and in particular the changes in the cross repression of transcription factors

Reviewer #2 (Remarks to the Author):

Many transcription factors have been shown to regulate oligodendrocyte differentiation and myelination; how they relate to each other and how extracellular signals influence them remains comparatively unclear however. Here, Weider and colleagues use a combination of enhancer analysis, NFATC inhibitors and conditional ablation of calcineurin in the oligodendrocyte lineage to show that Olig2 and Nkx2-2 mutually antagonize Sox10's activity

at each other's enhancer regions. This cross-antagonism between Olig2 and Nkx2-2 was already appreciated; the enhancer analysis gives a better understanding of the genetic basis for it. In luciferase assays, co-expression of constitutively active calcineurin A relieves this repression, raising the possibility that NFATC activity could allow co-expression of Olig2 and Nkx2-2 and promote oligodendrocyte differentiation. Consistent with this proposed role, they find that pharmacological inhibition of NFATC activation reduces oligodendrocyte differentiation in vitro and conditional ablation of calcineurin B1 reduces oligodendrocyte differentiation in vivo.

Overall these results should be of substantial interest to the myelin and myelin repair fields. As the authors note, involvement of the NFATCs would provide an important molecular mechanism to link extrinsic signals with the transcriptional program of oligodendrocyte differentiation and myelination. Although the extrinsic signals that may induce NFATC activation are not investigated, the manuscript will serve as an important basis for future work. Several aspects of the paper would benefit from revision prior to publication, however:

- 1) The cumulative data that the calcineurin/NFATC pathway influences oligodendrocyte differentiation are quite good. What is less clear is that it necessarily does so via induction of Nkx2-2. Presumably the NFATC proteins could regulate differentiation via other or additional targets? Further evidence that Nkx2-2 expression is downstream of NFATC activation and that this mediates the differentiation effect should be provided. For example, is Nkx2-2 down-regulated in VIVIT treated cultures? Can the effects of VIVIT or FK506 on differentiation be rescued by forced expression of Nkx2-2?
- 2) Reduced oligodendrocyte differentiation is shown with conditional ablation of CnB1 (which results in a ~50% reduction in the density of Mbp, Plp1, Myrf and Nkx2-2 positive cells in the spinal cord at E18.5). Due to the early time-point assessed, it is not clear whether this represents a transient delay in oligodendrocyte differentiation or a more substantial block that would ultimately cause a strong hypomyelination phenotype. Is it feasible to assess the spinal cords or brains of these mice postnatally to demonstrate a lasting effect on myelination (rather than early OL differentiation), or does the peripheral loss of CnB1 cause early death? If the latter is the case, it should be commented on.
- 3) The evidence that the NFATC proteins are differentially activated during oligodendrocyte differentiation is somewhat scant (Fig S3B shows some nuclear translocation of NFATC2 in differentiating conditions, though this is not quantified). Is it feasible to perform Western blots on NFATC1, 2 and 3 on oligodendrocyte cultures in proliferative and differentiating conditions to assess expression levels of phosphorylation states of the three presumably relevant NFATCs? Better still would be evidence of activation at the time of oligodendrocyte differentiation in vivo.

Minor points:

-The manuscript would benefit from some proof reading for typographic errors. (e.g., figure 1B should read 'ECR45' rather than 'ECR54').

- Figure 3B is low magnification; the subcellular localization of each NFATC is difficult to appreciate.

Reviewer #3 (Remarks to the Author):

The manuscript describes novel studies of the Nfat family of transcription factors in oligodendrocyte development. There is a network of oligodendrocyte transcription factors, most of which are functionally connected to Sox10, which is required for oligodendrocyte differentiation. While many of these transcription factors have been investigated, it is clear that Sox10 partners with many transcription factors in different contexts. One unresolved issue is that Sox10 can support activation of Nkx2.2 and Olig2, but these transcription factors themselves are mutually antagonistic. Studies in this manuscript identify Sox10-responsive enhancers in Nkx2.2 and Olig2 genes, and they find that sox10 activation can be augmented on these enhancers by members of the Nfat family of transcription factors. It is proposed based on transfection studies that Nfat activation can neutralize the cross-repression observed with Nkx2 on an Olig2 enhancer, and conversely, Olig2 repression of an Nkx2.2 enhancer.

In general, the experiments are of high quality, with some exceptions noted below, and elucidation of Nfat regulation in the oligodendrocyte development is a novel finding. Nonetheless, there are several limitations of the study in its current form. First, while the elucidation of enhancer activity is solid, most of the Nfat effects are relatively modest (2-fold), and the mechanistic analysis does not really distinguish between simple synergistic activation of Nfat with Sox10 vs. the proposed neutralization of inhibitory effects by either Nkx2.2 or Olig2. Given the magnitude of the effects, it would be reasonable to simply test if olig2 or Nkx2.2 affects protein levels of Sox10, and whether this is altered by Nfat expression. Another control would be to show that Nkx2.2 or Olig2 do not affect other Sox10-dependent enhancers.

Second, while it is proposed that Nfat proteins would limit cross-inhibition by Nkx2.2 or Olig2, this idea is not explicitly tested in a cellular experiment. For example, in Figure 2, it is shown that Sox10 expression does not drive simultaneous expression of Nkx2.2 and Olig2, and a similar experiment altering Nfat activation would be a better test of the proposed model.

Third, the targeted cre-driven deletion of Cnb1 is an important experiment and shows a ~2-fold reduction in mature oligodendrocytes. Based on Figure 6M, it is concluded that there is no change in proliferation to account for the altered number of oligodendrocytes in this experiment. However, there does seem to be an apparent decrease in Ki67 staining. While this is not significant, it probably means that this deserves a more careful quantitation (with $n > 3$) to rule out a proliferative change. Moreover, it should be appreciated that a "snapshot" measure of proliferation at one time point cannot rule out changes in proliferative rate over the time course of oligodendrocyte development that would reduce the total number of oligodendrocytes, as is observed in the absence of Cnb1. Accordingly,

the VIVIT treatment in Supp. Fig 5B suggests impaired proliferation but the increased variability probably requires >3 biological replicates.

Finally, the human oligodendrocyte experiments in Figure 7 are interesting, but there is concern that effects are observed only with near toxic levels of VIVIT (0.5 uM, when it is toxic >1 uM). This experiment needs an independent (pharmacological or siRNA) means to test the role of Nfat family members, if it were to be included in the manuscript.

Minor comments

The use of previously ChIP-seq data is helpful, but the authors may also comment on the status of active enhancer modifications (H3K27ac, also from Yu et al)

Figure 3B: it appears that all cells express Nfatc3, about half express Nfatc2, and none express Nfatc1. Perhaps the DAPI channel is obscuring the signal. The cytoplasmic staining is not apparent at the presented scale. The panels and description thereof needs to be revised.

3C: appears to be missing label for the 2nd bar in this figure

3E: The diagram suggest that the Sox10 site labeled dProm is not in the actual promoter of the upstream transcript.

Figure 5: Although the control values are set to 1, the presentation of the data suggests that there is essentially no variability under control conditions in panels B,C, E, F. If these experiments were done in pairwise fashion, that should be indicated, but otherwise, the variability of control cultures should be faithfully represented.

p. 8 "restitution" should presumably be "restoration"

"compensate" should be "compensate for"

Point by point response to the reviewers' comments
on manuscript NCOMMS-17-22147-T
by Weider et al.

" Nfat/calcineurin signaling promotes oligodendrocyte differentiation and myelination in health and disease by transcription factor network tuning "

We were very pleased with the very interested and supportive response by the referees and their constructive criticism. We carefully considered the reviewers' comments and addressed them as follows:

Reviewer #1:

This is an interesting paper that describes a role for calcineurin signalling in oligodendrocyte differentiation by the activation of Nfat. It also shows that Nfat exerts its effects by altering the repressive interactions between the sox10, olig2, and nkx2.2 transcription factor network. The demonstration that this mechanism is also active in human cells, albeit with a smaller effect size, adds importance, as does the work showing a nfat expression in MS lesions. Overall the paper will be of significant interest to neuroscientists.

Comment 1: A problem, though, is the clarity. It is noteworthy that the abstract and start of the discussion described the work by starting with nfat and calcineurin, and then move on to the work on the enhancer binding sites. In contrast, the results describe the work in exactly the opposite way. Personally, I found the logic of the results hard to follow and think that the paper would be much easier to read and have more scientific impact if the results were ordered in the same way as the other sections of the paper.

If this is not done, then the logic for switching from nkx2 .2 enhancers on page 6 and the selection of nfat on page 7 ("caught our attention" is hardly scientific) needs much better explanation.

Response 1: We followed the reviewer's suggestion and rearranged the results in the same order they are mentioned in abstract and start of discussion. We agree that this rearrangement has improved the logic.

Comment 2: How significant is it to find four peaks for transcription factor binding enhancers (page 5). What's the comparator here?

Response 2: We think that the four peaks are highly significant. Among 15 transcription factors with oligodendroglial expression, we found only 5 that exhibited more than two peaks, and among those 5 genes only Nkx2.2 and its close relative Nkx6.2 had multiple peaks for both Sox10 and Olig2. We added this information in the revised version as new Supl. Fig. 4b and describe and discuss our findings on p.10, top of the revised manuscript.

Comment 3: The experiments in figure 4B and C don't demonstrate joint activation. The additive effect on nkx2.2 expression could simply reflect different cell populations being recruited.

Response 3: We agree with the reviewer that this is formally possible. However, we do not think that this is the case because (i) electroporation rates were comparable for the co-electroporation and the single electroporations, and because $89.5 \pm 1.3\%$ ($p \leq 0.001$, $n = 3$) of all electroporated cells in the co-electroporations were double positive for Sox10 and Nfatc2. This information has been added on p.12, bottom of the revised manuscript.

Comment 4: Do the authors show that activated calcineurin increases nuclear localisation of nfatc2? This is an important experiment.

Response 4: To address this question we used ionomycin to activate calcineurins in primary rodent oligodendrocytes. We then investigated in two separate approaches whether nuclear localization of Nfatc2 would be increased. First, we co-transfected cells with a Nfatc2-YFP and a H2B-mCherry fusion protein and determined the YFP fraction that colocalized with the constitutively nuclear mCherry. Second, we determined signal intensities for nuclear and cytosolic endogenous Nfatc2 after immunostaining and calculated the quotient. Both approaches confirmed increased nuclear presence of Nfatc2 after ionomycin treatment. These results are presented as novel Fig. 3c,d and are described and discussed on p.7, bottom of the revised manuscript.

Comment 5: I suggest a figure summarising the pathway proposed and in particular the changes in the cross repression of transcription factors

Response 5: As suggested, we have added a summary as novel Fig. 7o to the revised version of the manuscript.

Reviewer #2

Many transcription factors have been shown to regulate oligodendrocyte differentiation and myelination; how they relate to each other and how extracellular signals influence them remains comparatively unclear however. Here, Weider and colleagues use a combination of enhancer analysis, NFATC inhibitors and conditional ablation of calcineurin in the oligodendrocyte lineage to show that Olig2 and Nkx2-2 mutually antagonize Sox10's activity at each other's enhancer regions. This cross-antagonism between Olig2 and Nkx2-2 was already appreciated; the enhancer analysis gives a better understanding of the genetic basis for it. In luciferase assays, co-expression of constitutively active calcineurin A relieves this repression, raising the possibility that NFATC activity could allow co-expression of Olig2 and Nkx2-2 and promote oligodendrocyte differentiation. Consistent with this proposed role, they find that pharmacological inhibition of NFATC activation reduces oligodendrocyte differentiation in vitro and conditional ablation of calcineurin B1 reduces oligodendrocyte differentiation in vivo.

Overall these results should be of substantial interest to the myelin and myelin repair fields. As the authors note, involvement of the NFATCs would provide an important molecular mechanism to link extrinsic signals with the transcriptional program of oligodendrocyte differentiation and myelination. Although the extrinsic signals that may induce NFATC activation are not investigated, the manuscript will serve as an important basis for future work. Several aspects of the paper would benefit from revision prior to publication, however:

Comment 6: The cumulative data that the calcineurin/NFATC pathway influences oligodendrocyte differentiation are quite good. What is less clear is that it necessarily does so via induction of Nkx2-2. Presumably the NFATC proteins could regulate differentiation via other or additional targets? Further evidence that Nkx2-2 expression is downstream of NFATC activation and that this mediates the differentiation effect should be provided. For example, is Nkx2-2 down-regulated in VIVIT treated cultures? Can the effects of VIVIT or FK506 on differentiation be rescued by forced expression of Nkx2-2?

Response 6: We agree with the reviewer that the original version of our manuscript was in need of additional evidence for Nkx2.2 as an important target of Nfat activity. We added several experiments to the revised version to provide such evidence.

A: Normally, Nkx2.2 expression is strongly upregulated during the first 24 hours of oligodendrocyte differentiation in culture. However, when differentiation is in the presence of VIVIT, this upregulation is substantially blunted arguing that Nfat activity is required for full Nkx2.2 induction during the early phases of differentiation. These data have been added as novel Fig. 4a to the revised manuscript and is described and discussed on p.9, bottom.

B: As suggested by the reviewer, we have also analyzed for the revised version whether forced Nkx2.2 expression in FK506-treated oligodendrocytes would rescue the differentiation defects in cultured oligodendrocytes. Using Mbp expression as readout, we were indeed able to show that Mbp expression levels in FK506-treated cultures returned back to normal when Nkx2.2 was supplied by retroviral transduction. These data have been added as novel Fig. 6g to the revised manuscript and are described and discussed on p.15, bottom half.

C: We also show that Nfatc2 has little impact on regulatory regions of myelin genes and thereby exclude an obvious alternative for Nfat function. These data are added as novel Supl. Fig. 4a and are described on p.9, top of the revised manuscript.

Comment 7: Reduced oligodendrocyte differentiation is shown with conditional ablation of CnB1 (which results in a ~50% reduction in the density of Mbp, Plp1, Myrf and Nkx2-2 positive cells in the spinal cord at E18.5). Due to the early time-point assessed, it is not clear whether this represents a transient delay in oligodendrocyte differentiation or a more substantial block that would ultimately cause a strong hypomyelination phenotype. Is it feasible to assess the spinal cords or brains of these mice postnatally to demonstrate a lasting effect on myelination (rather than early OL differentiation), or does the peripheral loss of CnB1 cause early death? If the latter is the case, it should be commented on.

Response 7: Indeed, it would have been nice to study the effects of calcineurin/Nfat signaling on myelination at later time points postnatally. However, as already guessed by the reviewer perinatal death occurs in our conditional mouse mutants because of peripheral CnB1 loss. Defects likely concern several neural crest derivatives. This early death precludes a postnatal analysis. We added this information to the revised manuscript on p.6 as third paragraph.

Comment 8: The evidence that the NFATC proteins are differentially activated during oligodendrocyte differentiation is somewhat scant (Fig S3B shows some nuclear translocation of NFATC2 in differentiating conditions, though this is not quantified). Is it feasible to perform Western blots on NFATC1, 2 and 3 on oligodendrocyte cultures in proliferative and

differentiating conditions to assess expression levels of phosphorylation states of the three presumably relevant NFATCs? Better still would be evidence of activation at the time of oligodendrocyte differentiation in vivo.

Response 8: To provide additional evidence for the activation of Nfat proteins during oligodendrocyte differentiation, the reviewer suggested to study phosphorylation states of Nfat proteins by Western blotting. There are a limited number of commercially available antibodies for such studies. Many of them are poorly characterized and none of the ones we checked worked in our hands on Western blots. We therefore chose alternative approaches:

A: We transfected the oligodendroglial cell line Oli-neu with NFATc2-YFP and H2B-mCherry fusion proteins, induced differentiation by dibutyryl-cAMP treatment and determined nuclear amounts of the Nfatc2-YFP fusion during the first days of differentiation. We found higher amounts of the Nfatc2-YFP fusion in the nucleus. Such increased nuclear localization is an indicator of increased activity. We added these data as novel Suppl. Fig. 3d and describe the results on p.7, bottom of the revised manuscript.

B: We show by quantification of immunocytochemical stainings of primary rat oligodendroglial cells that the amount of endogenous Nfatc2 also increases during the differentiation process. Again this increased nuclear localization should be an indicator of increased activity. These data are added as novel Suppl. Fig. 3c to the revised manuscript and described on p.7 bottom.

C: We also transduced oligodendroglial cultures with a tdTomato construct under control of multimerized Nfat binding sites and analyzed the location of this construct. We found that the tdTomato localized preferentially to differentiating Mbp-positive cells and at significantly higher rates than a tdTomato whose expression was not under Nfat control. This direct evidence for Nfat activity in differentiating oligodendrocytes was added as novel Suppl. Fig. 1e to the revised manuscript and is described on p.5, middle.

Comment 9 (minor): The manuscript would benefit from some proof reading for typographic errors. (e.g., figure 1B should read 'ECR45' rather than 'ECR54').

Response 9: We proofread the manuscript and corrected typographic errors including the one mentioned.

Comment 10 (minor): Figure 3B is low magnification; the subcellular localization of each NFATC is difficult to appreciate.

Response 10: We replaced the old panels by new high magnification pictures from a confocal microscope. The new Fig. 3b demonstrates the nuclear localization of Nfatc1, Nfatc2 and Nfatc3.

Reviewer #3

The manuscript describes novel studies of the Nfat family of transcription factors in oligodendrocyte development. There is a network of oligodendrocyte transcription factors, most of which are functionally connected to Sox10, which is required for oligodendrocyte differentiation. While many of these transcription factors have been investigated, it is clear that Sox10 partners with many transcription factors in different contexts. One unresolved issue is that Sox10 can support activation of Nkx2.2 and Olig2, but these transcription factors themselves are mutually antagonistic. Studies in this manuscript identify Sox10-responsive enhancers in Nkx2.2 and Olig2 genes, and they find that sox10 activation can be augmented on these enhancers by members of the Nfat family of transcription factors. It is proposed based on transfection studies that Nfat activation can neutralize the cross-repression observed with Nkx2 on an Olig2 enhancer, and conversely, Olig2 repression of an Nkx2.2 enhancer.

In general, the experiments are of high quality, with some exceptions noted below, and elucidation of Nfat regulation in the oligodendrocyte development is a novel finding. Nonetheless, there are several limitations of the study in its current form.

Comment 11: While the elucidation of enhancer activity is solid, most of the Nfat effects are relatively modest (2-fold), and the mechanistic analysis does not really distinguish between simple synergistic activation of Nfat with Sox10 vs. the proposed neutralization of inhibitory effects by either Nkx2.2 or Olig2. Given the magnitude of the effects, it would be reasonable to simply test if olig2 or Nkx2.2 affects protein levels of Sox10, and whether this is altered by Nfat expression. Another control would be to show that Nkx2.2 or Olig2 do not affect other Sox10-dependent enhancers.

Response 11: As suggested, we analyzed whether Olig2 or Nkx2.2 would reduce Sox10 levels and whether this reduction is counteracted by Nfatc2. In our experiments, we found no evidence that Olig2 or Nkx2.2 reduces endogenous Sox10 levels in oligodendroglial cells or its amounts in transfected cells. We also failed to detect a positive influence of Nfatc2 on Sox10 levels. These results have been included as novel Suppl. Fig. 6f-h and are described and discussed on p. 14, bottom and p.15, top.

Comment 12: While it is proposed that Nfat proteins would limit cross-inhibition by Nkx2.2 or Olig2, this idea is not explicitly tested in a cellular experiment. For example, in Figure 2, it is shown that Sox10 expression does not drive simultaneous expression of Nkx2.2 and Olig2, and a similar experiment altering Nfat activation would be a better test of the proposed model.

Response 12: The suggested experiment would have required the generation and characterization of new transgenic mice. Generation of a tet-inducible Nfatc2 transgene and combination with the Cre and tTA allele alone would have taken a year. Therefore, we chose a different approach. To further substantiate our model of relieved cross-inhibition between Nkx2.2 and Olig2, we extended our chicken electroporation experiments and studied whether co-electroporation of Sox10 and Nfatc2 would increase the number of cells that simultaneously expressed Nkx2.2 and Olig2 as compared to electroporations of single transcription factors. In support of our model, this was indeed the case. We have added these data as novel Fig. 6f and discuss them on p.15, second paragraph of the revised manuscript.

Comment 13: The targeted cre-driven deletion of Cnb1 is an important experiment and shows a ~2-fold reduction in mature oligodendrocytes. Based on Figure 6M, it is concluded that there is no change in proliferation to account for the altered number of oligodendrocytes in this experiment. However, there does seem to be an apparent decrease in Ki67 staining. While this is not significant, it probably means that this deserves a more careful quantitation (with n>3) to rule out a proliferative change. Moreover, it should be appreciated that a “snapshot” measure of proliferation at one time point cannot rule out changes in proliferative rate over the time course of oligodendrocyte development that would reduce the total number of oligodendrocytes, as is observed in the absence of Cnb1. Accordingly, the VIVIT treatment in Supp. Fig 5B suggests impaired proliferation but the increased variability probably requires >3 biological replicates.

Response: We performed additional Ki67 stainings on more Sox10::Cre CnB1 knockouts and in VIVIT-treated cultures. Even with n = 6, we failed to detect a statistically relevant decrease in cell proliferation *in vivo*. In cell culture, there was a very mild decrease in proliferation rate (n = 7). However, this decrease cannot account for the much bigger effect on oligodendroglial differentiation. We also like to point out that several other data throughout the manuscript (including the ones obtained for the Cnp1::Cre driven CnB1 deletion) show that the calcineurin/Nfat effect is in the early phase of differentiation and not in OPCs making it additionally unlikely that the effect is mainly one on proliferation. We have replaced the old

Fig. 6M and Suppl. Fig. 5B (now. Fig. 2m and Suppl. Fig. 1b) with new panels in which the number of biological replicates was increased.

Comment 14: The human oligodendrocyte experiments in Figure 7 are interesting, but there is concern that effects are observed only with near toxic levels of VIVIT (0.5 μ M, when it is toxic $>1 \mu$ M). This experiment needs an independent (pharmacological or siRNA) means to test the role of Nfat family members, if it were to be included in the manuscript.

Response: We have to apologize for not being clearer in the original version. It did not become clear from our wording that only 5 μ M but not 1 μ M of VIVIT is toxic for human iOL. This is 10 times higher than the concentration we used. In the revised version, we have added data on iOL viability at various VIVIT concentrations. We also determined the total iOL number after 21 days of treatment with 0.5 μ M VIVIT and thereby demonstrate that the reduction of O4 positive cells under these conditions is not due to cell death. These data are shown as novel Suppl. Fig. 7a,b and described on p.16, second paragraph of the revised manuscript.

As suggested by the referee, we also treated human iOL with FK506 as a second calcineurin inhibitor. We first determined toxicity of FK506 for iOL and then used a non-toxic concentration to show that FK506 inhibits the differentiation of human iOL into O4 positive as well as MBP positive oligodendrocytes at day 21 and 35 respectively, very similar to the effect of VIVIT. This effect is based on the inhibition of early stages of oligodendroglial differentiation as shown by the lack of a FK506 effect on the number of MBP positive cells over O4 when added from day 21 to 35. This new set of data including controls has been added as novel Fig. 7h-j and Suppl. Fig. 7e,f to the revised manuscript and is discussed on p. 16, second paragraph.

Comment 15 (minor): The use of previously ChIP-seq data is helpful, but the authors may also comment on the status of active enhancer modifications (H3K27ac, also from Yu et al)

Response 15: We added the information on p.10, top of the revised manuscript.

Comment 16 (minor): Figure 3B: it appears that all cells express Nfatc3, about half express Nfatc2, and none express Nfatc1. Perhaps the DAPI channel is obscuring the signal. The cytoplasmic staining is not apparent at the presented scale. The panels and description thereof needs to be revised.

Response 16: This comment is similar to comment 10 of reviewer 2. As described above we exchanged the panels by new ones of higher magnification and quality.

Comment 17 (minor): Figure 3C: appears to be missing label for the 2nd bar in this figure

Response 17: The label was checked. The three “-“ indicate that the value is from Oln93 cells that lack endogenous Sox10 because of CRISPR/Cas9-mediated inactivation (first “-“), and that were not transduced by control (second “-“) or Sox10-expressing lentivirus (third “-“).

Comment 18 (minor): Figure 3E: The diagram suggest that the Sox10 site labeled dProm is not in the actual promoter of the upstream transcript.

Response 18: The diagram was modified to show that the dProm fragment used in our study is indeed in the actual promoter of the upstream transcript.

Comment 19 (minor): Figure 5: Although the control values are set to 1, the presentation of the data suggests that there is essentially no variability under control conditions in panels B,C, E, F. If these experiments were done in pairwise fashion, that should be indicated, but otherwise, the variability of control cultures should be faithfully represented.

Response 19: We are sorry that we did not state that more clearly. The experiments were compared in a pairwise fashion. This is now described in the Figure legend.

Comment 20 (minor): p. 8 “restitution” should presumably be “restoration” “compensate” should be “compensate for”

Response 20: This was corrected.

REVIEWERS' COMMENTS:

Reviewer #1 (Remarks to the Author):

This revised ms addresses the points in my first review in a satisfactory way. I do not think however that the re-ordering has improved the ms as the start is now very abrupt. Surely the logical first set of expts would be the determination of nfat expression in oligodendrocytes, after explaining why they are good candidates for regulation of differentiation? Then, expts manipulating their function would make more sense. If this cant be done, it might be better to go back to the original order, even though this is different to the abstract, as this seems more comfortable to the authors.

Reviewer #2 (Remarks to the Author):

In this revised manuscript, Weider et al. provide substantial new evidence to show that NFAT proteins modulate oligodendrocyte differentiation. In particular, they now use several additional methods (immunohistochemistry, fusion proteins and transfection of reporter constructs) to better demonstrate that NFATCs (NFATC2 in particular) are activated during oligodendrocyte differentiation. They also provide additional experimentation to show that the effects of calcineurin inhibition can be rescued by forced expression of Nkx2-2, confirming it as an important downstream target and mediator of the NFATCs. The manuscript has been somewhat re-ordered, improving its logical flow.

Overall, the concerns I raised for the first submission have been well-addressed. This is now an excellent manuscript that identifies NFATs as an important link between intracellular signaling and the transcriptional regulation of oligodendrocyte differentiation and myelination.

Reviewer #3 (Remarks to the Author):

The revised manuscript has adequately addressed the concerns raised in the previous round of review, and it is now suitable for publication

The only exception is that Suppl. Figure 6 f,g shows 3-10 fold variability in Sox10 protein levels from experiment to experiment that probably reflects varying levels of transfection efficiency. The consequent variability does not allow a conclusive answer to the specific question posed here, and therefore the data should be deleted.

Point by point response to the reviewers' comments
on manuscript NCOMMS-17-22147A
by Weider et al.

" Nfat/calcineurin signaling promotes oligodendrocyte differentiation and myelination by transcription factor network tuning "

We were very pleased with the response of the reviewers and carefully considered the remaining comments as follows:

Reviewer #1:

This revised ms addresses the points in my first review in a satisfactory way. I do not think however that the re-ordering has improved the ms as the start is now very abrupt. Surely the logical first set of expts would be the determination of nfat expression in oligodendrocytes, after explaining why they are good candidates for regulation of differentiation? Then, expts manipulating their function would make more sense. If this cant be done, it might be better to go back to the original order, even though this is different to the abstract, as this seems more comfortable to the authors.

Response:

We thank the reviewer for acknowledging the improvements made during the revision. As part of the first revision, we had followed the reviewer's request and restructured the Results section in such a way that the order of experiments matched the order in which they are mentioned in the abstract. While we agree that this increased readability of the manuscript, we do not think that the renewed suggestion to rewrite would do the same. In our manuscript, we first show that Nfat activity influences oligodendrocyte differentiation before going into the

molecular details and determining which Nfat protein occurs and how it functions on a molecular level. The reviewer would prefer us to present the molecular details on Nfats first and then provide evidence for their relevance in oligodendrocyte differentiation. To us these two approaches appear equally valid. Which one to choose rather seems a matter of personal taste. Reviewer #2, for instance, explicitly commends us for the structure of the revised version. As we cannot see a clear advantage in the suggested re-ordering, we chose to stick by the one of the revised version.

Reviewer #2:

In this revised manuscript, Weider et al. provide substantial new evidence to show that NFAT proteins modulate oligodendrocyte differentiation. In particular, they now use several additional methods (immunohistochemistry, fusion proteins and transfection of reporter constructs) to better demonstrate that NFATCs (NFATC2 in particular) are activated during oligodendrocyte differentiation. They also provide additional experimentation to show that the effects of calcineurin inhibition can be rescued by forced expression of Nkx2-2, confirming it as an important downstream target and mediator of the NFATCs. The manuscript has been somewhat re-ordered, improving its logical flow.

Overall, the concerns I raised for the first submission have been well-addressed. This is now an excellent manuscript that identifies NFATs as an important link between intracellular signaling and the transcriptional regulation of oligodendrocyte differentiation and myelination.

Response:

We thank the reviewer for acknowledging the improvements made during the revision.

Reviewer #3:

The revised manuscript has adequately addressed the concerns raised in the previous round of review, and it is now suitable for publication.

The only exception is that Suppl. Figure 6 f,g shows 3-10 fold variability in Sox10 protein levels from experiment to experiment that probably reflects varying levels of transfection efficiency. The consequent variability does not allow a conclusive answer to the specific question posed here, and therefore the data should be deleted.

Response:

We thank the reviewer for acknowledging the improvements made during the revision. We also agree with the reviewer's assessment that the variability in Suppl. Fig. 6f,g is quite high. As presumed by the reviewer, variability is indeed caused by varying transfection efficiency and inherent to this kind of assay. To account for the variability, we repeated the experiment nine times. Although a lower variability would be desirable, we still think the experiment worth showing as it refutes correlations between the presence of a particular transcription factor and the Sox10 amount. Therefore we chose to leave the data in the manuscript, but mention the caveat on p.14, bottom of the final revised manuscript.